# DH-Fusion: Depth-Aware Hybrid Feature Fusion for Multimodal 3D Object Detection

## Abstract

State-of-the-art LiDAR-camera 3D object detectors usually focus on feature fusion. However, they neglect the factor of depth while designing the fusion strategy. In this work, we for the first time point out that different modalities play different roles as depth varies via statistical analysis and visualization. Based on this finding, we propose a Depth-Aware Hybrid Feature Fusion (DH-Fusion) strategy that guides the weights of point cloud and RGB image modalities by introducing depth encoding at both global and local levels. Specifically, the Depth-Aware Global Feature Fusion (DGF) module adaptively adjusts the weights of image Bird's-Eye-View (BEV) features in multi-modal global features via depth encoding. Furthermore, to compensate for the information lost when transferring raw features to the BEV space, we propose a Depth-Aware Local Feature Fusion (DLF) module, which adaptively adjusts the weights of original voxel features and multi-view image features in multi-modal local features via depth encoding. Extensive experiments on the nuScenes and KITTI datasets demonstrate that our DH-Fusion method surpasses previous state-of-the-art methods. Moreover, our DH-Fusion is more robust to various kinds of corruptions, outperforming previous methods on nuScenes-C w.r.t. both NDS and mAP.

## 1 Introduction

3D object detection has a wide range of applications in the fields of autonomous driving and robotics. A large number of previous works have successfully focused on using a single modality, such as point cloud or images, to design efficient 3D object detectors. However, the performance of these detectors reaches a bottleneck due to the limitations of modality characteristics. For instance, the point cloud modality can only provide rich geometric information while lacks detailed semantic information; the image modality can only provide rich texture information while lacks three-dimensional spatial information. To address the aforementioned issues, we are highly motivated to obtain comprehensive information that represents objects by designing a LiDAR-camera 3D object detector.

In recent years, LiDAR-camera 3D object detection develops rapidly. Some works Liu et al. (2023); Liang et al. (2022); Bai et al. (2022); Cai et al. (2023); Yin et al. (2024) propose effective methods to integrate information from two modalities at the feature level. However, they all overlook an important factor of depth in their fusion strategies. To understand how point cloud and image information vary with depth, we first conduct statistical and visualization analysis on the nuScenes-mini dataset Caesar et al. (2020), and find that: (1) The number of points representing objects at near range is relatively large, which allows us to accurately determine the object's location, size, and category, even without the aid of images. As shown in Fig. 1a, there is an average of 163.7 points per object within 0-10 meters, which is a substantial number. We also visualize a car at 6.8 meters in Fig. 1b ① and find it encompasses a considerable number of points, well representing the shape. In contrast, some background noise in the image may interfere with detection (Fig. 1b ②). (2) As the depth increases, the number of points representing objects decreases rapidly. As shown in Fig. 1a, the number of points within 30-50 meters falls below one per object, meaning that many objects are even not represented by any points, such as the object at 42.1 meters in Fig. 1b ③. In contrast, the complete objects may still be observed on the image, as in Fig. 1b ④, where the image information becomes more important. To address the above problems, we propose a feature fusion strategy that adaptively adjusts the importance of the two modalities based on depth.

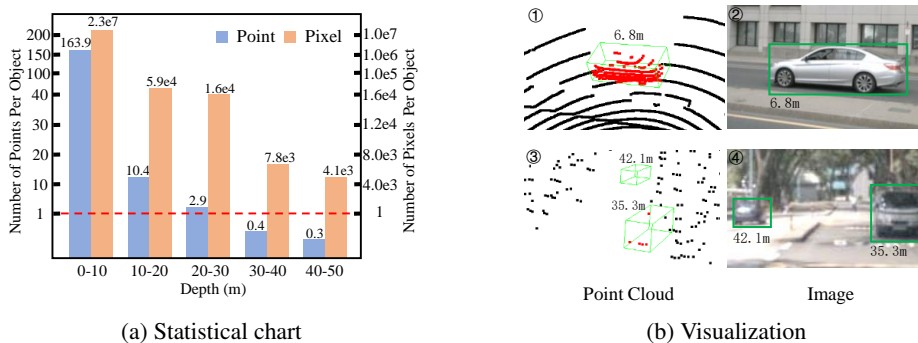

(a) Statistical chart  (b) Visualization

Figure 1: Statistical and visualization analysis on the nuScenes-mini dataset. (a) The average numbers of points and pixels for each object at different depths. (b) Examples of near-range and long-range objects in images and point cloud. Points within the bounding boxes are colored red for observation.

Specifically, we propose a novel method for multi-modal 3D object detection, namely Depth-Aware Hybrid Feature Fusion (DH-Fusion). The innovation lies in adaptively adjusting the weights of features by introducing depth encoding to hybrid feature fusion at both global and local levels. The fusion strategy consists of two crucial components: Depth-Aware Global Feature Fusion (DGF) module and Depth-Aware Local Feature Fusion (DLF) module. In DGF, we take point cloud Bird's-Eye-View (BEV) features and image BEV features as inputs, and dynamically adjust the weights of image BEV features based on depth during fusion by utilizing a global-fusion transformer encoder with a depth encoder. To compensate for the information lost when transforming raw features to BEV space, we enhance the fused BEV features at a lower cost by utilizing the original instance features. In DLF, we obtain 3D boxes by utilizing a Region Proposal Network (RPN). Then, the 3D boxes are projected into both LiDAR voxel features and multi-view image features to crop out corresponding local instance features with more detailed information. Afterward, we take these as inputs and dynamically adjust the weights of local multi-view image features and local LiDAR voxel features based on depth through the use of a local-fusion transformer encoder with the depth encoder. In the end, we update local features for each object on the global feature map to enhance the detailed instance information of multi-modal global features for detection.

Our contributions are summarized as follows.

1. We for the first time point out that depth is an important factor to consider while fusing LiDAR point cloud features and RGB image features for 3D object detection. From our statistical and visualization analysis, we can see that image features play different roles as depth varies.

2. We propose a depth-aware hybrid feature fusion strategy that dynamically adjusts the weights of features during feature fusion by introducing depth encoding at both global and local levels. The above strategy can obtain high-quality features for detection, fully leveraging the advantages of different modalities at various depths.

3. Our method is evaluated on the nuScenes Caesar et al. (2020) dataset, KITTI Geiger et al. (2012) dataset, and a more challenging nuScenes-C Dong et al. (2023) dataset, outperforming previous multi-modal methods and being robust to various kinds of data corruptions.

## 2  RELATED WORK

Since our method is based on conducting 3D object detection using data from multiple modalities, including point cloud and images, we briefly review recent works in the following fields: LiDAR-based 3D object detection, camera-based 3D object detection, and LiDAR-camera 3D object detection.

## 2.1 LiDAR-based 3D Object Detection

LiDAR-based 3D object detectors only take the point cloud as input. Based on their different data representations, they can be divided into point-based Yang et al. (2018; 2019); Shi et al. (2019); Shi & Rajkumar (2020); Shi et al. (2020b), voxel-based Yan et al. (2018); Zhou & Tuzel (2018); Lang et al. (2019); Deng et al. (2021); Yin et al. (2021a), and point-voxel-based Shi et al. (2020a; 2022); Hu et al. (2022) methods. The feature extraction networks of point-based methods typically extract features directly from the point cloud through a point-based backbone Qi et al. (2017), such as PointRCNN Shi et al. (2019). The voxel-based methods first convert the point cloud into voxels and then extract voxel features through a 3D sparse convolution network Graham et al. (2018), such as VoxelNet Zhou & Tuzel (2018). Point-voxel-based methods like PV-RCNN Shi et al. (2020a) combine the above two methods to extract and fuse point and voxel features. The purpose of these approaches is to capture the geometric spatial information of the point cloud. However, point cloud is sparse and incomplete, lacking detailed texture information, which greatly limits the detection performance.

## 2.2 Camera-based 3D Object Detection

Camera-based 3D object detectors only take images as inputs. Depending on the form of inputs, they can be divided into monocular Liu et al. (2020); Li et al. (2019a); Brazil & Liu (2019); Qin et al. (2019); Shi et al. (2021); Wang et al. (2021b), stereo Li et al. (2019b); Chen et al. (2020); You et al. (2019); Sun et al. (2020); Liu et al. (2021a), and multi-view Wang et al. (2022); Huang et al. (2021); Li et al. (2022c); Yang et al. (2023) 3D object detectors. Early works like FCOS3D Wang et al. (2021b) input a monocular image and utilize 2D object detectors to directly predict 3D bounding boxes, but these approaches have limited capability in capturing spatial information. Subsequently, stereo and multi-view 3D object detectors are proposed to obtain more precise depth information by constructing spatial relationships among multiple images, such as Stereo RCNN Li et al. (2019b) and BEVDet Huang et al. (2021). These methods successfully achieve purely visual 3D object detection, but they do not perform as well as LiDAR-based methods, because the spatial depth information provided by images is not as direct and precise as that provided by point cloud.

## 2.3 LiDAR-Camera 3D Object Detection

LiDAR-camera 3D object detectors take point cloud and images as inputs, and can be classified into early-fusion-based Vora et al. (2020); Wang et al. (2021a); Xu et al. (2021); Yin et al. (2021b); Wu et al. (2023), intermediate-fusion-based Liu et al. (2023); Liang et al. (2022); Bai et al. (2022); Cai et al. (2023); Yin et al. (2024), and late-fusion-based Pang et al. (2020; 2022) 3D object detectors based on the location of multi-modal information fusion Mao et al. (2023).

Early-fusion-based methods perform at the point level, where the typical approach involves enhancing the raw point cloud with semantic information extracted from images. PointPainting Vora et al. (2020) and FusionPainting Xu et al. (2021) decorate the raw point cloud with semantic scores from 2D semantic segmentation. Similarly, PointAugmenting Wang et al. (2021a) enhances the raw point cloud using features extracted from a 2D semantic segmentation network. However, early-fusion-based methods are sensitive to alignment errors between the two modalities.

Intermediate-fusion-based methods perform at the feature level. Transfusion Bai et al. (2022) first proposes to utilize the transformer for fine-grained fusion from LiDAR BEV features and multi-view image features. FUTR3D Chen et al. (2023a) encode each modality using deformable attention Zhu et al. (2020) in its own coordinate and concatenate them for fusion. BEVFusion Liang et al. (2022); Liu et al. (2023) projects both point cloud and images to BEV space for BEV feature fusion. SparseFusion Xie et al. (2023) extracts instance-level features from both two modalities separately, and fuse them to perform detection. Similarly, ObjectFusion Cai et al. (2023) utilizes 3D proposals from LiDAR modality to extract instance-level features for fusion. CMT Yan et al. (2023) proposes the simultaneous interaction between the object queries and multi-modal features in the transformer encoder and decoder. LoGoNet Li et al. (2023) and IS-Fusion Yin et al. (2024) propose feature fusion at both the instance level and scene level. The intermediate-fusion-based methods gradually become a mainstream approach due to the diversity of fusion strategies.

Figure 2: Overview of our method. It introduces depth encoding in both global and local feature fusion to obtain depth-adaptive multi-modal representations for detection. $\otimes$ is the multiplication operation, and Ⓜ is the merge operation.

Late-fusion-based methods perform at the bounding box level. Typically, CLOCs Pang et al. (2020) obtains 2D and 3D bounding boxes by separately using 2D and 3D object detectors, and then combine them to achieve more accurate 3D bounding boxes. However, the interaction between modalities in late-fusion-based methods is very limited, which constrains model performance.

These multi-modal methods successfully outperform single-modal methods. However, their feature fusion methods do not take depth into account. In contrast, our approach introduces depth information to guide the hybrid feature fusion, boosting the performance of the detector.

## 3 METHODOLOGY

In this section, we first give an overview of our proposed multi-modal 3D object detector, and then provide a detailed introduction to our proposed feature fusion method.

### 3.1 OVERVIEW

We propose a multi-modal 3D object detection method via Depth-Aware Hybrid Feature Fusion (DH-Fusion). As illustrated in Fig. 7, our approach consists of two important feature fusion modules: Depth-Aware Global Feature Fusion (DGF) and Depth-Aware Local Feature Fusion (DLF). In the following, we briefly describe the detection pipeline.

**Inputs.** First, we take the point cloud $P$ and multi-view images $I$ as inputs, where point cloud consists of a set of points: $P = \{P_1, P_2, \cdots, P_{N_l}\}$, and each point has four dimensions: X-axis, Y-axis, Z-axis, and intensity; the multi-view images comprise $N_c$ images: $I = \{I_1, I_2, \cdots, I_{N_c}\}$, each image captured by its corresponding camera.

**Input Encoding.** For the point cloud $P$, we use a 3D encoder to extract raw global voxel features $\mathcal{V}_O^G$; for the multi-view images $I$, we use a 2D encoder to extract image features of all views $\mathcal{I}_O^G$.

**Hybrid Feature Fusion.** Then, for voxel features $\mathcal{V}_O^G$, we compress the height dimension to obtain point cloud BEV features $\mathcal{V}_B^G$; for image features $\mathcal{I}_O^G$, we transform their perspective view to bird's eye view to obtain image BEV features $\mathcal{I}_B^G$. To fully leverage the features from two modalities, we design a DGF module that aims to dynamically adjust the weights of image BEV features based on depth values during feature fusion. Please refer to Sec. 3.2 for more details. To compensate for the information lost when transforming raw features to BEV space, we propose a DLF module that, based on depth, utilizes the raw features to enhance the detailed information of each object instance in global multi-modal features. It consists of three processes: local feature selection, local feature fusion, and merging local features into global features. First, we obtain the local multi-modal BEV features $\mathcal{F}_B^L$, local voxel features $\mathcal{V}_O^L$, and local multi-view image features $\mathcal{I}_O^L$, by cropping the corresponding global features based on the 3D boxes obtained from an RPN; then, it dynamically and individually adjusts the weights of each local feature of $\mathcal{V}_O^L$ and $\mathcal{I}_O^L$ based on depth values during

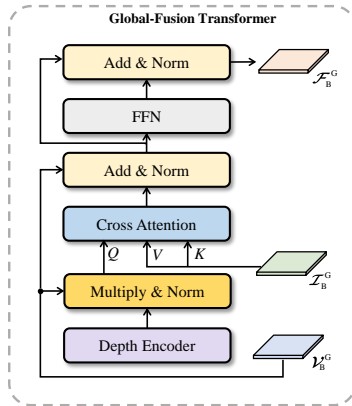 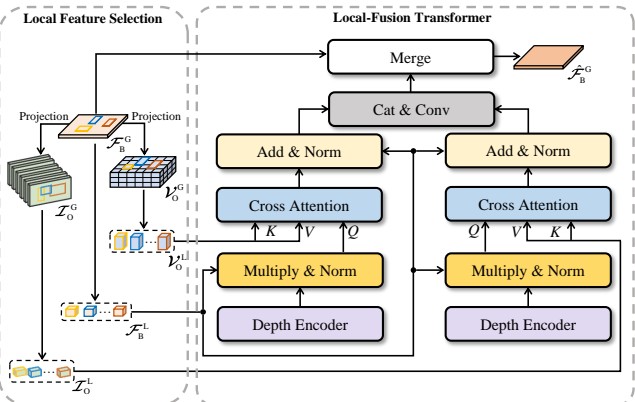

Figure 3: Illustration of the DGF. It consists of a global fusion transformer with the depth encoder.

Figure 4: Illustration of the DLF. It consists of a local feature selection module and a local fusion transformer with the depth encoder.

feature fusion; finally, we update local features for each object on the global feature map. Please refer to Sec. 3.3 for more details. In this way, we obtain enhanced multi-modal global features for detection.

**Decoding.** Based on the enhanced multi-modal global features $\hat{\mathcal{F}}_B^G$ that contain rich semantic and spatial information, we utilize a transformer decoder and a detection head to predict the object categories and 3D bounding boxes.

## 3.2 DEPTH-AWARE GLOBAL FEATURE FUSION

As shown in Fig. 3, the DGF module consists of a global-fusion transformer with a depth encoder. In the following, we provide a detailed explanation of each component.

### 3.2.1 DEPTH ENCODER

We introduce depth encoding (DE) in feature fusion to dynamically adjust the weights of image BEV features during fusion. First, we build a depth matrix $M$ to store the depth value of each position element $p_k$ represented as:

$$p_k = \{(x_k, y_k) : d_k\}, k \in [1, n], \tag{1}$$

where $(x_k, y_k)$ are the positional coordinates, $d_k$ is the depth value, and $n$ is the number of elements. Then, we use Euclidean distance to calculate the distance between every element's spatial location $(x_k, y_k)$ and the ego coordinate element's location $(x_{\frac{n}{2}}, y_{\frac{n}{2}})$:

$$d_k = E((x_k, y_k), (x_{\frac{n}{2}}, y_{\frac{n}{2}})), k \in [1, n], \tag{2}$$

where we denote $E(\cdot)$ as the Euclidean distance calculation. The depth matrix $M$ serves as a lookup table to avoid redundant computation of depth values. Since the size of the BEV features is large and the depth distribution is simple, to avoid introducing additional parameters, the depth encoding $De$ is obtained by applying sine and cosine functions Vaswani et al. (2017) to the depth matrix.

### 3.2.2 GLOBAL-FUSION TRANSFORMER

In the global-fusion transformer, we take the point cloud BEV features $\mathcal{V}_B^G \in \mathbb{R}^{W \times H \times C}$ and image BEV features $\mathcal{I}_B^G \in \mathbb{R}^{W \times H \times C}$ as inputs, and integrate the depth encoding obtained above by multiplying it with the point cloud BEV features, forming the query $Q_{\mathcal{V}}^G = N(\mathcal{V}_B^G \times Conv(De))$, where $Conv(\cdot)$ is a convolution operation to align with the channels of $\mathcal{V}_B^G$, and $N(\cdot)$ is a normalization

layer. The image BEV features are queried as the corresponding key $K_\mathcal{I}^G$ and value $V_\mathcal{I}^G$. We utilize the multi-head cross attention to achieve the interacted feature $\hat{\mathcal{V}}_B^G$ based on depth:

$$\hat{\mathcal{V}}_B^G = CA(Q_\mathcal{V}^G, K_\mathcal{I}^G, V_\mathcal{I}^G), \tag{3}$$

where $CA(\cdot)$ indicates the multi-head cross attention. Afterward, we aggregate the information from both modalities to obtain the fused features $\mathcal{F}_B^G$:

$$\mathcal{F}_B^G = N(FFN(N(\hat{\mathcal{V}}_B^G + \mathcal{V}_B^G))) + N(\hat{\mathcal{V}}_B^G + \mathcal{V}_B^G), \tag{4}$$

where $N(\cdot)$ is a normalization layer; $FFN(\cdot)$ specifies a feed-forward network containing two convolution operations. In this way, we obtain fused features in which the image features play different roles as the depth varies.

### 3.3 DEPTH-AWARE LOCAL FEATURE FUSION

As shown in Fig. 4, the DLF module consists of a local feature selection and a local-fusion transformer with the depth encoder. In the following, we provide a detailed explanation of each component.

#### 3.3.1 LOCAL FEATURE SELECTION

To compensate for the information lost when transforming point cloud features and image features to BEV space, we enhance the instance details of fused BEV features $\mathcal{F}_B^G$ using instance features from raw voxel features $\mathcal{V}_O^G$ and multi-view image features $\mathcal{I}_O^G$. Specifically, we utilize an RPN to regress $t$ 3D boxes based on the BEV features $\mathcal{F}_B^G$. We directly crop the global fused BEV features $\mathcal{F}_B^G$ based on the regressed 3D boxes to obtain the local fused BEV features $\mathcal{F}_B^L \in \mathbb{R}^{c \times t}$. On the other hand, we project the 3D boxes onto the raw voxel features and multi-view image features to obtain their corresponding local features before global fusion, preserving richer information for each object instance. Specifically, we utilize the voxel pooling operation Deng et al. (2021), followed by a 3D convolution operation and a linear layer, to extract local voxel features $\mathcal{V}_O^L \in \mathbb{R}^{c \times t}$; we transform the 3D boxes from bird's eye view to perspective view, and utilize the RoI Align operation He et al. (2017), followed by a linear layer, to extract instance image features $\mathcal{I}_O^L \in \mathbb{R}^{c \times t}$. By doing this, we obtain the hybrid (before & after global fusion) local features, which will be sent to the subsequent fusion module.

#### 3.3.2 LOCAL-FUSION TRANSFORMER

In the local-fusion transformer, the weights of each local raw feature are dynamically adjusted based on depth values during feature fusion, and we update local features for each object on the global feature map. Specifically, we take the local multi-modal BEV features $\mathcal{F}_B^L$, local voxel features $\mathcal{V}_O^L$, and local multi-view image features $\mathcal{I}_O^L$ as inputs, and integrate the depth encoding by multiplying it with the local multi-modal BEV features, forming the query $Q_\mathcal{F}^L$. The local multi-view image features and local voxel features are respectively queried as the corresponding key $K_\mathcal{I}^L$, $K_\mathcal{V}^L$ and value $V_\mathcal{I}^L$, $V_\mathcal{V}^L$. The two multi-head cross-attention modules are utilized to achieve the interacted features $\hat{Q}_\mathcal{F}^L, \hat{Q}_\mathcal{F}^{L'}$. Note that the computation process of multi-head cross attention is similar to that described in Sec. 3.2.2 and is omitted here. Afterward, we aggregate the above features:

$$\hat{\mathcal{F}}_B^L = Conv(Cat(\hat{Q}_\mathcal{F}^L + \mathcal{F}_B^L, \hat{Q}_\mathcal{F}^{L'} + \mathcal{F}_B^{L'})), \tag{5}$$

where $Cat(\cdot)$ is the concatenation operation; $Conv(\cdot)$ is used to align with the feature channels of global fused BEV features $\mathcal{F}_B^G$. As a result, we obtain enhanced local features by dynamically calling back rich information in raw modalities at various depths. Afterward, we update the global features $\mathcal{F}_B^G$ by inserting the enhanced local features at corresponding locations.

## 4 EXPERIMENTS

In this section, we will first introduce the dataset and evaluation metrics, followed by the implementation details. Then, we will compare our method with the state-of-the-art methods on nuScenes and KITTI datasets, and also present results on a more challenging dataset of nuScenes-C with data corruptions. Finally, we will show the ablation studies and qualitative results. More comparison experiments and ablation studies are provided in Appendix A.3 and A.4.

### 4.1 EXPERIMENTAL SETUP

**Datasets and evaluation metrics.** We evaluate our proposed DH-Fusion on the nuScenes Caesar et al. (2020), KITTI Geiger et al. (2012) datasets, and a more challenging dataset of nuScenes-C Dong et al. (2023) with data corruptions. The nuScenes dataset provides 700 scene sequences for training, 150 scene sequences for validation, and 150 scene sequences for testing. The KITTI dataset contains 7481 samples for training and 7518 samples for testing. We split the original training set into training and validation sets, following Shi et al. (2020a). The nuScenes-C dataset provides 27 corruptions with 5 severities on the nuScenes validation set, including corruptions at the weather, sensor, motion, object, and alignment level. For the nuScenes and nuScenes-C datasets, we use the nuScenes detection scores (NDS) and mean Average Precision (mAP) to evaluate our detection results, where NDS is a comprehensive metric that combines object translation, scale, orientation, velocity, and attribute errors. For the KITTI dataset, we use 3D Average Precision (3D AP) with recall 40 positions to evaluate our detection results.

**Implementation details.** We implement the proposed DH-Fusion with PyTorch Paszke et al. (2019) under the open-source framework MMDetection3D Contributors (2020). Specifically, on the nuScenes dataset, for the LiDAR branch, we use VoxelNet Zhou & Tuzel (2018) with FPN Yan et al. (2018) as the 3D encoder. The voxel size is set to [0.075m, 0.075m, 0.1m], and the range of point cloud is [-54m, 54m] along the X-axis, [-54m, 54m] along the Y-axis, and [-3m, 5m] along the Z-axis. For the image branch, we use the ResNet18 He et al. (2016), ResNet50 He et al. (2016), and SwinTiny Liu et al. (2022) with FPN Lin et al. (2017) as the 2D image encoder of DH-Fusion-light, -base, -large, respectively. Correspondingly, the resolution of input images is resized to $256 \times 704$, $320 \times 800$, and $384 \times 1056$. On the KITTI dataset, for the LiDAR branch, the voxel size is set to [0.05m, 0.05m, 0.1m], and the range of point cloud is [0, 70.4m] along the X-axis, [-40m, 40m] along the Y-axis, and [-3m, 1m] along the Z-axis. For the image branch, we follow Li et al. (2023) to use SwinTiny with FPN as the 2D image encoder, and the resolution of input images is resized to $187 \times 621$. Additionally, we utilize BEVPoolV2 Huang & Huang (2022) to obtain image BEV features. Following Liu et al. (2023), the feature size $W \times H$ is set to $180 \times 180$, the channel $C$ is set to 128, and the channel $c$ is also set to 128. The multi-head cross attention is implemented with 8 heads, and the FFN contains 2 MLP layers with a hidden dimension of 128. Following Xie et al. (2023), the number of regressed 3D boxes $t$ is set to 200. More implementation details are provided in Appendix A.1.

### 4.2 COMPARISONS ON THE NUSCENES DATASET

Aiming for a fair comparison, we categorize previous methods based on the types of 2D backbones into ResNet50-based, SwinTiny-based, and others, and provide four versions of our proposed method, named DH-Fusion-light, DH-Fusion-base, DH-Fusion-large, and DH-Fusion-large+. The comparison results are shown in Tab. 1, and the detailed results with more metrics are in Appendix A.5.1. (1) Compared with the ResNet50-based methods, our DH-Fusion-base outperforms the top method FocalFormer3D Chen et al. (2023b) by up to 1 pp w.r.t. NDS under the same configuration. Specifically, we reach 74.0% w.r.t. NDS and 71.2% w.r.t. mAP on the validation set, and 74.7% w.r.t. NDS and 71.7% w.r.t. mAP on the test set, while maintaining comparable inference speed of 8.7 FPS on a 3090 GPU. (2) Moreover, our DH-Fusion-light surpasses the typical BEVFusion Liu et al. (2023) by up to 1 pp w.r.t. all metrics using a lighter 2D backbone, and achieves a real-time inference speed of 13.8 FPS. (3) Compared with the SwinTiny-based methods and others, our DH-Fusion-large outperforms the top method IS-Fusion Yin et al. (2024) under the same configuration, and runs 2x faster than it. In addition, our DH-Fusion-large+ with a larger image size and 2D backbone surpasses all previous SOTA methods. Specifically, we reach 74.9% w.r.t. NDS on the validation set, and 75.8% w.r.t. NDS on the test set, while achieving the inference speed of 3.4 FPS

Table 1: Comparisons with the state of the art on the nuScenes `validation` and `test` sets. FPS is measured on a 3090 GPU by default, and * denotes the inference speed on an A100 GPU referred from the original paper. Note that all results are obtained without any model ensemble or test time augmentation.

| Methods | Present at | Image Size - 2D Backbone | FPS | Validation NDS | Validation mAP | Test NDS | Test mAP |
|---|---|---|---|---|---|---|---|
| Image Backbone: ResNet50He et al. (2016) | | | | | | | |
| Trainsfusion Bai et al. (2022) | CVPR'22 | 320 × 800-ResNet50 | 6.5 | 71.3 | 67.5 | 71.7 | 68.9 |
| DeepInteraction Yang et al. (2022) | NeurIPS'22 | 448 × 800-ResNet50 | 1.9 | 72.4 | 69.9 | 73.4 | 70.8 |
| MSMDFusion Jiao et al. (2023) | CVPR'23 | 448 × 800- ResNet50 | 2.1 | 72.1 | 69.7 | 74.0 | 71.5 |
| FocalFormer3D Chen et al. (2023b) | ICCV'23 | 320 × 800-ResNet50 | 9.2* | 73.1 | 70.1 | 73.9 | 71.6 |
| **DH-Fusion-base (Ours)** | - | 320 × 800-ResNet50 | **8.7** | **74.0** | **71.2** | **74.7** | **71.7** |
| Image Backbone: SwinTinyLiu et al. (2021b) | | | | | | | |
| BEVFusion Liang et al. (2022) | NeurIPS'22 | 448 × 800-SwinTiny | 0.7* | 71.0 | 67.9 | 71.8 | 69.2 |
| BEVFusion Liu et al. (2023) | ICRA'23 | 256 × 704- SwinTiny | 9.6 | 71.4 | 68.5 | 72.9 | 70.2 |
| ObjectFusion Cai et al. (2023) | ICCV'23 | 256 × 704- SwinTiny | - | 72.3 | 69.8 | 73.3 | 71.0 |
| SparseFusion Xie et al. (2023) | ICCV'23 | 256 × 704- SwinTiny | 4.4 | 72.8 | 70.5 | 73.8 | 72.0 |
| IS-Fusion Yin et al. (2024) | CVPR'24 | 384 × 1056-SwinTiny | 3.2* | 74.0 | 72.8 | 75.2 | 73.0 |
| GAFusion Li et al. (2024) | CVPR'24 | 448 × 800-SwinTiny | - | 73.5 | 72.1 | 74.9 | 73.6 |
| Image Backbone: Others | | | | | | | |
| AutoAlignV2 Chen et al. (2022b) | ECCV'22 | 640 × 1280-CSPNet Wang et al. (2020) | 4.8* | 71.2 | 67.1 | 72.4 | 68.4 |
| UVTR Li et al. (2022a) | NeurIPS'22 | 640 × 1280-ResNet101 He et al. (2016) | 1.8 | 70.2 | 65.4 | 71.1 | 67.1 |
| FUTR3D Chen et al. (2023a) | CVPR'23 | 900 × 1600-VOVNet Lee et al. (2019) | 3.3* | 68.0 | 64.2 | 72.1 | 69.4 |
| UniTR Wang et al. (2023b) | ICCV'23 | 256 × 704-DSVT Wang et al. (2023a) | 9.3* | 73.3 | 70.5 | 74.5 | 70.9 |
| CMT Yan et al. (2023) | ICCV'23 | 640 × 1600-VOVNet | 6.0* | 72.9 | 70.3 | 74.1 | 72.0 |
| UniPAD Yang et al. (2024) | CVPR'24 | 900 × 1600-ConvNeXtS Liu et al. (2022) | - | 73.2 | 69.9 | 73.9 | 71.0 |
| SparseLIF Zhang et al. (2024) | ECCV'24 | 900 × 1600-VOVNet | - | 74.6 | 71.2 | - | - |
| **DH-Fusion-light (Ours)** | - | 256 × 704-ResNet18 | **13.8** | 73.3 | 69.8 | 74.2 | 70.9 |
| **DH-Fusion-large (Ours)** | - | 384 × 1056-SwinTiny | 5.7 | 74.4 | 72.3 | 75.4 | 72.8 |
| **DH-Fusion-large+ (Ours)** | - | 900 × 1600-ConvNeXtS | 3.4 | **74.9** | **72.9** | **75.8** | **73.6** |

Table 2: Comparisons with the state of the art on the KITTI *val* set. The results are on **3D AP** with IoU=0.7, 0.5, 0.5 for three classes: Car, Pedestrian, and Cyclist. We use bold for the best results, and underline for the second best results.

| Methods | Car Easy | Car Mod. | Car Hard | Car mAP | Pedestrian Easy | Pedestrian Mod. | Pedestrian Hard | Pedestrian mAP | Cyclist Easy | Cyclist Mod. | Cyclist Hard | Cyclist mAP | mAP |
|---|---|---|---|---|---|---|---|---|---|---|---|---|---|
| PointFusion Xu et al. (2018) | 77.9 | 63.0 | 53.3 | 64.7 | 33.4 | 28.0 | 23.4 | 28.3 | 49.3 | 29.4 | 27.0 | 35.3 | 42.8 |
| F-PointNet Qi et al. (2018) | 83.8 | 70.9 | 63.7 | 72.8 | 70.0 | 61.3 | 53.6 | 61.6 | 77.2 | 56.5 | 53.4 | 62.3 | 65.6 |
| CLOCs Pang et al. (2020) | 89.5 | 79.3 | 77.4 | 82.1 | 62.9 | 56.2 | 50.1 | 56.4 | 87.6 | 67.9 | 63.7 | 73.1 | 70.5 |
| 3D-CVF Yoo et al. (2020) | 89.7 | 79.9 | 78.5 | 82.7 | - | - | - | - | - | - | - | - | - |
| EPNet Huang et al. (2020) | 88.8 | 78.7 | 78.3 | 81.9 | 66.7 | 59.3 | 54.8 | 60.3 | 83.9 | 65.6 | 62.7 | 70.7 | 71.0 |
| FocalsConv Chen et al. (2022a) | 92.3 | 85.3 | 83.0 | 86.8 | - | - | - | - | - | - | - | - | - |
| CAT-Det Zhang et al. (2022) | 90.1 | 81.5 | 79.2 | 83.6 | 74.1 | 66.4 | 58.9 | 66.5 | 87.6 | 72.8 | 68.2 | 76.2 | 75.4 |
| VFF Li et al. (2022b) | 92.3 | 85.5 | 82.9 | 86.9 | 73.3 | 65.1 | 60.0 | 66.1 | 89.4 | 73.1 | 69.9 | 77.5 | 76.9 |
| LoGoNet Li et al. (2023) | 92.0 | 85.0 | 84.3 | 87.1 | 70.2 | 63.7 | 59.5 | 64.5 | 91.7 | 75.4 | 72.4 | 79.8 | 77.1 |
| VirConv Wu et al. (2023) | 94.9 | 90.0 | 88.1 | 91.0 | 73.3 | 66.9 | 60.4 | 66.9 | 90.0 | 73.9 | 69.1 | 77.7 | 78.5 |
| **DH-Fusion (Ours)** | 95.6 | 89.6 | 87.7 | 91.0 | 75.6 | 67.5 | 63.7 | 68.9 | 91.8 | 74.5 | 69.8 | 78.7 | 79.5 |

on a 3090 GPU. Overall, our method achieves higher detection accuracy and faster inference speed.

## 4.3 COMPARISONS ON THE KITTI DATASET

To further evaluate the effectiveness of our proposed method, we evaluate our DH-Fusion on the KITTI dataset. Tab. 2 shows the results on the KITTI *val* set. Compared with previous multi-modal methods, DH-Fusion achieves state-of-the-art mAP performance, surpassing the top method VirConv Wu et al. (2023) by 1 pp. The results demonstrate the effectiveness of our method on the KITTI dataset. Notably, for the small-sized pedestrian class, our DH-Fusion achieves 63.7% w.r.t. AP at the hard difficulty level and 68.9% w.r.t. mAP, outperforming VirConv by up to 3 pp. This indicates that our method is also effective for small-sized objects, while they are at a distant range or extremely occluded. Overall, these results further validate the effectiveness of DH-Fusion.

## 4.4 ROBUSTNESS TO CORRUPTIONS

We further implement some experiments on the nuScenes-C Dong et al. (2023) dataset to evaluate the model's robustness under various corruptions, including changes in weather, data loss or temporal-spatial misalignment in multi-modal inputs, etc. The results for different kinds of corrup-

Table 3: Robustness experiments on nuScenes-C. Numbers are **NDS / mAP**.

| Methods | None | Corruption | | | | | Average |
|---|---|---|---|---|---|---|---|
| | | Weather | Sensor | Motion | Object | Alignment | |
| FUTR3D Chen et al. (2023a) | 68.05 / 64.17 | 62.75 / 55.51 | 63.66 / 56.83 | 53.16 / 44.43 | 65.45 / 61.04 | 62.83 / 57.60 | $62.82^{\downarrow5.23}$ / $56.99^{\downarrow7.18}$ |
| TransFusion Bai et al. (2022) | 69.82 / 66.38 | 65.42 / 59.37 | 66.17 / 59.82 | 51.52 / 41.47 | 68.28 / 64.38 | 61.98 / 54.94 | $63.74^{\downarrow6.08}$ / $58.73^{\downarrow7.65}$ |
| BEVFusion Liu et al. (2023) | 71.40 / 68.45 | 67.54 / 61.87 | 67.59 / 61.80 | 55.19 / 47.30 | 68.01 / 65.14 | 63.94 / 58.71 | $66.06^{\downarrow5.34}$ / $61.03^{\downarrow7.42}$ |
| **DH-Fusion-light (Ours)** | **73.30 / 69.75** | **72.19 / 67.48** | **69.16 / 62.87** | **57.07 / 47.52** | **71.01 / 67.11** | **67.24 / 62.38** | $\mathbf{68.67}^{\downarrow4.63}$ / $\mathbf{63.07}^{\downarrow6.68}$ |

Table 4: Ablation studies of each proposed module.

| Baseline | DGF | DLF | NDS | mAP |
|---|---|---|---|---|
| ✓ | | | 71.4 | 68.5 |
| ✓ | ✓ | | $72.4^{\uparrow1.0}$ | $69.4^{\uparrow0.9}$ |
| ✓ | | ✓ | $72.7^{\uparrow1.3}$ | $69.3^{\uparrow0.8}$ |
| ✓ | ✓ | ✓ | $\mathbf{73.3}^{\uparrow1.9}$ | $\mathbf{69.8}^{\uparrow1.3}$ |

Table 5: Ablation studies of depth encoding (DE) in DGF and DLF.

| Methods | NDS | mAP |
|---|---|---|
| Baseline + DGF | 72.4 | 69.4 |
| w/o DE | $71.8^{\downarrow0.6}$ | $69.0^{\downarrow0.4}$ |
| Baseline + DLF | 72.7 | 69.3 |
| w/o DE | $71.6^{\downarrow1.1}$ | $68.4^{\downarrow0.9}$ |
| DH-Fusion-base | 74.0 | 71.2 |
| w/o DE | $73.5^{\downarrow0.5}$ | $70.0^{\downarrow1.2}$ |

Table 6: Ablation studies of different operations for depth encoding.

| Methods | NDS | mAP |
|---|---|---|
| Summation | 72.8 | 69.2 |
| Concatenation | 72.5 | 68.7 |
| Multiplication | **73.3** | **69.8** |

tions are shown in Tab. 3, and more detailed results for each fine-grained corruption are shown in Appendix A.5.2. We find that our DH-Fusion-light still achieves an average performance of 68.67% w.r.t. NDS and 63.07% w.r.t. mAP under various corruptions, which only decreases by 4.63 pp w.r.t. NDS and 6.68 pp w.r.t. mAP, compared to its performance without corruptions. Performance drop is smaller than that observed with previous methods including BEVFusion Liang et al. (2022) across all kinds of corruptions, indicating that our DH-Fusion-light possesses superior robustness. Furthermore, we observe that our DH-Fusion-light is particularly robust against weather and object corruptions, where the performance drop is less than 3pp. The more stable performance indicates that our method is more friendly to practical applications, where data corruption may occur.

## 4.5 ABLATION STUDIES

We conduct ablation studies to first demonstrate the effect of each component of DH-Fusion, then to demonstrate the effect of depth encoding in DGF and DLF, and finally to assess the impact of multiplying depth encoding. All method variants are implemented on the nuScenes validation dataset.

**Effect of DGF and DLF.** To demonstrate the effect of DGF and DLF, we conduct experiments by integrating the components one by one into the baseline, BEVFusion Liu et al. (2023). The results are shown in Tab. 4. We find that our DGF improves the baseline performance by 1.0 pp w.r.t. NDS and 0.9 pp w.r.t. mAP. This demonstrates that dynamically adjusting the weights of the image BEV features during fusion is effective for 3D object detection. Additionally, our DLF improves the baseline performance by 1.3 pp w.r.t. NDS and 0.8 pp w.r.t. mAP, which indicates that dynamically adjusting the weights of the local raw instance features based on depth during fusion effectively compensates for the information loss caused by the transformation of global features into the BEV feature space. The results of integrating both components show an improvement of 1.9 pp w.r.t. NDS and 1.3 pp w.r.t. mAP, well verifying the benefits of dynamically fusing global and local hybrid features based on depth.

**Effect of depth encoding in DGF and DLF.** To evaluate the effectiveness of our depth encoding, we conduct experiments where the depth encoding is removed from the DGF and DLF modules, respectively. The results are shown in Tab. 5. When removing the depth encoding from Baseline+DGF, the performance drops by 0.6 pp w.r.t. NDS and 0.4 pp w.r.t. mAP. Similarly, when removing the depth encoding from Baseline+DLF, the performance also decreases by 1.1 pp w.r.t. NDS and 0.9 pp w.r.t. mAP. Additionally, when removing the depth encoding of DGF and DLF from DH-Fusion-base, the performance also decreases by 0.5 pp w.r.t. NDS and 1.2 pp w.r.t. mAP. These results indicate that our depth encoding is effective. Furthermore, we observe that removing the depth encoding from the DLF module results in a larger performance drop, suggesting that depth encoding plays a more crucial role in local feature fusion. We analyze that local feature fusion theoretically relies on the fine-grained geometric details of objects to enhance feature representations. For instance, objects located at extreme distances often have sparse and incomplete point cloud representations, which lack sufficient fine-grained information. If the point cloud and image features of such an object are fused into the BEV representation with equal weights, this could degrade the

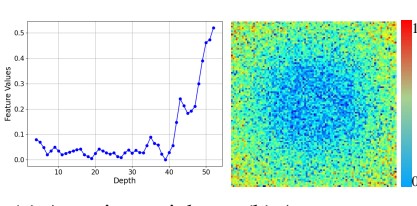

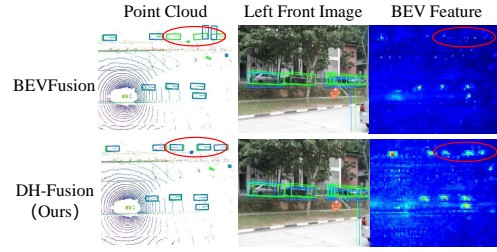

(a) Attention weights    (b) Average map

Figure 5: Attention weights applied on BEV image features in DGF vary with depth.

Figure 6: Qualitative detection results and BEV features of BEVFusion and ours. We show the ground truth boxes in green, and the prediction boxes in blue.

BEV features and badly affect detection performance. Our depth encoding addresses this issue by dynamically reducing the weight of incomplete point cloud features, thereby mitigating the negative impact of incomplete point cloud features on BEV features. Simultaneously, it integrates the fine-grained information from images into the BEV features, achieving feature enhancement.

**Impact of different operations for depth encoding.** We conduct experiments with different operations of depth encoding, including concatenation, summation, and multiplication. The results in Tab. 6, show that the multiplication operation consistently outperforms the summation and concatenation operations w.r.t. both metrics. The superior performance of multiplication can be attributed to its ability to more effectively modulate the feature maps based on depth information. Unlike summation, which simply shifts the feature values, or concatenation, which increases the dimensionality without direct interaction, multiplication allows for more interaction between the depth encoding and features, leading to better feature representation and ultimately improving the detection performance.

## 4.6 QUALITATIVE RESULTS

To better understand how depth encoding affects the feature fusion, in Fig. 5, we plot a curve to observe how the attention weights applied on the image BEV features in our DGF module vary with depth, and visualize the average attention map. It is evident that the weights of the image BEV features stay low in near range, but go up significantly as depth increases when the depth is larger than 40 meters. This trend supports our hypothesis that the image modality would become more important as depth increases. In this way, our depth encoding allows the model to dynamically adjust the weights of image BEV features based on depth.

We also compare the detection results of our DH-Fusion method with the baseline BEVFusion Liu et al. (2023) in Fig. 6, where we clearly find that our method better localizes those distant objects compared to BEVFusion. These results demonstrate that our proposed multi-modal fusion strategy based on depth is more effective for detection. Besides, we exhibit the corresponding BEV feature maps, where our method shows a stronger feature response for the foreground objects, especially for distant ones. That is why our feature fusion strategy can provide higher-quality detection results. More qualitative results can be found in Appendix A.6.

## 5 CONCLUSION

In this paper, we for the first time point out that different modalities play different roles as depth varies via statistical analysis and visualization. Based on this finding, we propose a feature fusion strategy for multi-modal 3D object detection, namely Depth-Aware Hybrid Feature Fusion (DH-Fusion), that dynamically adjusts the weights of features during feature fusion by introducing depth encoding at both global and local levels. Extensive experiments on the nuScenes and KITTI datasets demonstrate that our DH-Fusion method surpasses previous state-of-the-art methods. Moreover, our DH-Fusion is more robust to various kinds of corruptions, outperforming previous methods on the nuScenes-C dataset. We hope our method offers useful insights for feature fusion in the field of multi-modal 3D object detection.

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

# A APPENDIX

## A.1 ADDITIONAL IMPLEMENTATION DETAILS

During training, we adopt a one-stage strategy like DAL Huang et al. (2023). For the nuScenes dataset, the whole pipeline is trained for a total of 20 epochs with the AdamW optimizer Loshchilov & Hutter (2017) loading from the pre-trained weights from the ImageNet Deng et al. (2009) classification task only. Meanwhile, we use CBGS Zhu et al. (2019) to resample the training data, and the one-cycle learning policy with a maximum learning rate of $2.0 \times 10^{-4}$. The batch size is set to 2 per GPU on 4 3090 RTX GPUs. For the KITTI dataset, we follow all the configurations of Li et al. (2023) to train the whole model for 80 epochs. We adopt random flipping along both X and Y-axis, the random scaling in [0.95, 1.05], and random rotation in [-$\pi$/8, $\pi$/8] to augment the LiDAR data, and the random rotation in [-5.4°, 5.4°] and random resizing in [-0.06, 0.44] to augment the images. During evaluation, we test a single model without any data augmentation on a single 3090 RTX GPU.

## A.2 MORE ANALYSIS OF ROBUSTNESS OF DH-FUSION

We attribute the robustness of our methods to the combined effects of depth encoding. We analyze the reason for this using the foggy condition as an example: It is evident that fog has a greater impact on the image modality. Our depth-aware feature fusion adjusts the fusion weights based on the quality of the features. In such a case, the degradation of image feature quality leads to a reduction in the weight assigned to the image modality during fusion. As a result, the fused features with depth guidance are of higher quality than those without depth guidance. In this way, it reduces the negative impact of corruptions to some extent.

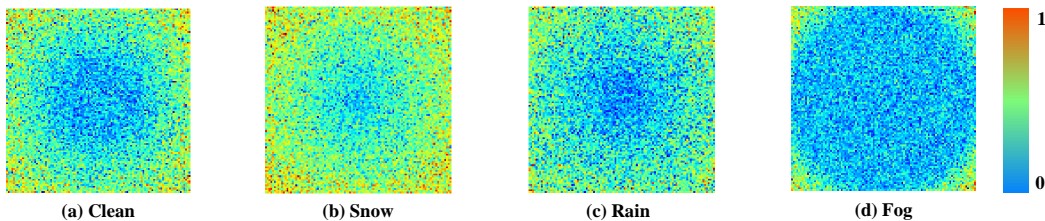

(a) Clean      (b) Snow      (c) Rain      (d) Fog

Figure 7: Average map of attention weights applied on BEV image features in DGF vary with depth under diverse weather conditions.

## A.3 ADDITIONAL COMPARISON EXPERIMENTS

### A.3.1 MODEL COMPUTATIONAL EFFICIENCY

We provide specific metrics on the computational efficiency of our method in Tab. 7. We attribute the speed improvement to our depth-aware feature fusion method. Due to the efficiency of our depth-aware feature fusion method, we can use a smaller backbone while maintaining competitive model performance. For example, compared to our baseline, BEVFusion (71.4% NDS and 68.5% mAP), our DH-Fusion-light achieves comparable performance (73.3% NDS and 69.8% mAP) using a smaller ResNet-18 backbone, while significantly improving inference speed (from 9.6 fps to 13.8 fps).

Table 7: Comparisons of parameter size, FLOPs, and latency.

| Methods | Image Size | 2D Backbone | NDS | mAP | Parameters (M) | FLOPs (G) | Latency (ms) |
|---------|-----------|-------------|-----|-----|----------------|-----------|--------------|
| BEVFusion | $256 \times 704$ | Swin-Tiny | 71.4 | 68.5 | 40.84 | 253.2 | 104.17 |
| BEVFusion + Ours | $256 \times 704$ | Swin-Tiny | 73.8 | 71.5 | 56.95 | 271.7 | 126.51 |
| DH-Fusion-tiny | $256 \times 704$ | ResNet-18 | 73.3 | 69.8 | 40.38 | 242.6 | 72.46 |
| DH-Fusion-base | $320 \times 800$ | ResNet-50 | 74.0 | 71.2 | 56.15 | 822.8 | 114.94 |
| DH-Fusion-large | $384 \times 1056$ | Swin-Tiny | 74.4 | 72.3 | 56.94 | 1508.2 | 175.44 |
| DH-Fusion-large+ | $900 \times 1600$ | ConvNeXt-S | 74.9 | 72.9 | 78.85 | 1973.8 | 294.12 |

### A.3.2  3D MULTI-OBJECT TRACKING EXPERIMENTS

We evaluate our DH-Fusion on the nuScenes tracking benchmark for 3D multi-object tracking (MOT) task. Following ObjectFusion Cai et al. (2023), we adopt the same tracking-by-detection algorithm that uses velocity-based closest point distance matching, which is more effective than 3D Kalman filter Chiu et al. (2020). For fair comparisons, we report the results of our DH-Fusion-light capable of real-time detection on the nuScenes validation set, as shown in Tab. 8. We find that our DH-Fusion-light outperforms BEVFusion Liu et al. (2023) and ObjectFusion Cai et al. (2023) by 2.0 pp and 0.6 pp w.r.t. AMOTA. These results demonstrate that our DH-Fusion provides 3D detection boxes of higher quality, benefiting the downstream task of 3D MOT.

Table 8: Comparisons on nuScenes validation set for 3D multi-object tracking.

| Methods | AMOTA ↑ | AMOTP ↓ | IDS ↓ |
|---------|---------|---------|-------|
| TransFusion Bai et al. (2022) | 71.8 | 60.3 | 694 |
| BEVFusion Liu et al. (2023) | 72.8 | 59.4 | 764 |
| ObjectFusion Cai et al. (2023) | 74.2 | 54.3 | 611 |
| **DH-Fusion-light (Ours)** | **74.8** | **50.3** | **539** |

### A.3.3  EVALUATION AT DIFFERENT DEPTHS

Since our fusion strategy is depth-aware, it is necessary to validate our method at different depths. Following Cai et al. (2023), we categorize annotation and prediction ego distances into three groups: Near (0-20m), Middle (20-30m), and Far (>30m). As shown in Tab. 9, compared to ObjectFusion Cai et al. (2023), our DH-Fusion-light consistently improves performance across all depth ranges. Specifically, our method achieves a 47.1 mAP at a far distance (>30m), surpassing ObjectFusion by 5.5 pp w.r.t. mAP. These results indicate that our method is more effective across different depths, especially in detecting distant objects.

Table 9: Comparisons on the nuScenes validation set at different depths. The numbers are **mAP**.

| Methods | Near | Middle | Far |
|---------|------|--------|-----|
| TransFusion-L Bai et al. (2022) | 77.5 | 60.9 | 34.8 |
| BEVFusion Liu et al. (2023) | 79.4 | 64.9 | 40.0 |
| ObjectFusion Cai et al. (2023) | 79.7 | 65.4 | 41.6 |
| **DH-Fusion-light (Ours)** | **80.3** | **66.5** | **47.1** |

### A.3.4  EVALUATION ON SMALL-SIZED OBJECTS

To further verify that our method is friendly to small-sized object detection, we conduct experiments on the nuScenes dataset to evaluate our method on normal-sized objects at a far distance and small-sized objects at a near distance. We consider cars as normal-sized objects, and pedestrians, motorcycles, and bicycles as small-sized objects here. As shown in Tab. 10, for these above small objects, our method outperforms the state-of-the-art method IS-Fusion, as well as our baseline BEV-Fusion. These results indicate that our depth-aware feature fusion benefits small object detection.

Table 10: Comparisons on small objects, including normal-sized objects at a far distance ($>30$m) and small-sized objects at a near distance (0-20m). The numbers are **AP**.

| Methods | >30m | 0-20m | | |
|---|---|---|---|---|
| | Car | Pedestrian | Motorcycle | Bicycle |
| BEVFusion Liu et al. (2023) | 72.1 | 92.9 | 89.9 | 75.7 |
| IS-Fusion Yin et al. (2024) | 76.1 | 94.1 | 90.2 | 78.4 |
| **DH-Fusion-large (Ours)** | **77.2** | **94.2** | **91.5** | **78.6** |

## A.4 ADDITIONAL ABLATION STUDIES

### A.4.1 EFFECT OF COSINE FUNCTIONS IN DEPTH ENCODER

To further evaluate the design of our depth encoder, we conduct the experiments using normalized depth directly as the depth encoding in our feature fusion module, without applying cosine functions. Our experimental results in Tab. 11 show a performance drop when using normalized depth directly. We argue that depth encoding benefits from the use of cosine functions to capture the periodicity and symmetry of the depth information relative to the ego vehicle. The cosine function helps in better representing the variations in depth, leading to model performance improvement.

Table 11: Ablation studies of cosine functions.

| Methods | NDS | mAP |
|---|---|---|
| BEVFusion + DGF | 72.4 | 69.4 |
| w/o cosine functions | $72.1^{\downarrow 0.3}$ | $68.5^{\downarrow 0.9}$ |
| BEVFusion + DLF | 72.7 | 69.3 |
| w/o cosine functions | $72.3^{\downarrow 0.4}$ | $68.6^{\downarrow 0.7}$ |

### A.4.2 EFFECT OF DEPTH ENCODER ON DIFFERENT BASELINES

To evaluate the generalization ability of our proposed depth encoder, we conduct an experiment using higher-performance IS-Fusion Yin et al. (2024) as the baseline, and integrate our depth encoder into its IGF module, which allows it to adjust the weights of image features with depth during instance feature fusion. We note that our method still achieves improvements. This demonstrates the generalization ability of our approach across different baselines.

Table 12: Ablation studies using IS-Fusion as the baseline.

| Methods | NDS | mAP |
|---|---|---|
| IS-Fusion Yin et al. (2024) | 73.6 | 72.5 |
| w/ DE | $74.1^{\uparrow 0.5}$ | $72.9^{\uparrow 0.4}$ |

### A.4.3 EFFECT OF DEPTH ENCODER

We attempt to improve depth encoding by setting it as a learnable matrix or incorporating a set of learnable parameters into the depth encoding. The results, as shown in the Tab. 13, indicate that these learnable or adaptive depth encoding methods do not lead to improved detection performance. Moreover, these methods undoubtedly introduce additional parameters to the model. We analyze that the learnable approaches may disrupt the true depth variations, making it difficult for the model to learn the correct weights based on the features.

## A.5 MORE DETAILED RESULTS

### A.5.1 DETAILED RESULTS ON THE NUSCENES VALIDATION AND TEST DATASETS

We further provide the detailed results of our method on the metrics mATE, mASE, mAOE, mAVE, and mAAE on the nuScenes validation and test dataset in Tab. 14 and 15. The results show the consistency improvements from our light version to the large+ version.

Table 13: Experiments of alternative depth encoding methods in DH-Fusion-tiny.

| Methods | NDS | mAP |
|---|---|---|
| unlearnable depth encoding | **73.3** | **69.8** |
| parameterized depth encoding | 71.8 | 67.8 |
| unlearnable depth encoding + learnable parameters | 72.7 | 69.5 |

Table 14: Detailed results on nuScenes `test` set.

| Methods | NDS ↑ | mAP ↑ | mATE ↓ | mASE ↓ | mAOE ↓ | mAVE ↓ | mAAE ↓ |
|---|---|---|---|---|---|---|---|
| DH-Fusion-light | 74.2 | 70.9 | 26.1 | 24.3 | 32.4 | 17.8 | **12.2** |
| DH-Fusion-base | 74.7 | 71.7 | 25.2 | 23.6 | 32.9 | 18.5 | 12.7 |
| DH-Fusion-large | 75.4 | 72.8 | 24.7 | 23.2 | 32.1 | 17.7 | 12.5 |
| DH-Fusion-large+ | **75.8** | **73.6** | **24.2** | **23.0** | **31.5** | **17.2** | 12.3 |

Table 15: Detailed results on nuScenes `validation` set.

| Methods | NDS ↑ | mAP ↑ | mATE ↓ | mASE ↓ | mAOE ↓ | mAVE ↓ | mAAE ↓ |
|---|---|---|---|---|---|---|---|
| DH-Fusion-light | 73.3 | 69.8 | 27.2 | 25.0 | 26.4 | 17.9 | 18.3 |
| DH-Fusion-base | 74.0 | 71.2 | 26.8 | 24.8 | 27.9 | 17.9 | 18.2 |
| DH-Fusion-large | 74.4 | 72.3 | 26.3 | 24.7 | 26.5 | 17.8 | 18.2 |
| DH-Fusion-large+ | **74.9** | **72.9** | **25.8** | **24.3** | **24.7** | 17.8 | **18.1** |

### A.5.2 DETAILED RESULTS ON THE NUSCENES-C

We further provide the detailed results of each fine-grained corruption on nuScenes-C in Tab. 16. The results are highly consistent with the average values of each kind of data corruption.

### A.6 MORE VISUALIZATION

As an extension of Fig. 6 in the manuscript, we provide additional examples of 3D object detection results and BEV features from our baseline, BEVFusion Liu et al. (2023), and our DH-Fusion. In various samples, our method consistently achieves higher accuracy and recall in 3D detection results, with stronger feature responses for distant objects compared to BEVFusion. These results demonstrate the effectiveness of the proposed method in dynamically adjusting the weights of features based on depth during fusion at both global and local levels.

### A.7 LIMITATION

The FOV Lost experiments in Tab. 16 highlight a potential limitation of our method: the establishment of a strong association in multi-modal feature fusion, which makes it sensitive to the loss of a modality.

Table 16: Comparisons for each corruption level on the nuScenes-C. Corruptions exist in both modalities by default. (L) means that only the point cloud modality has corruptions, and (C) means that only the image modality has corruptions. Numbers are **NDS / mAP**.

| Corruption | | FUTR3D | TransFusion | BEVFusion | **DH-Fusion** |
|---|---|---|---|---|---|
| None | | 68.5 / 64.17 | 69.82 / 66.38 | 71.40 / 68.45 | **73.30 / 69.75** |
| Weather | Snow | 61.52 / 52.73 | 68.29 / 63.30 | 68.33 / 62.84 | **71.47 / 65.98** |
| | Rain | 64.47 / 58.40 | 69.40 / 65.35 | 70.14 / 66.13 | **72.05 / 67.32** |
| | Fog | 61.20 / 53.19 | 62.62 / 53.67 | 62.73 / 54.10 | **72.13 / 67.24** |
| | Sunlight | 63.61 / 57.70 | 61.36 / 55.14 | 68.95 / 64.42 | **73.18 / 69.44** |
| Sensor | Density | 67.58 / 63.72 | 69.42 / 65.77 | 71.01 / 67.79 | **72.94 / 69.15** |
| | Cutout | 66.91 / 62.25 | 68.30 / 63.66 | 70.09 / 66.18 | **71.99 / 67.45** |
| | Crosstalk | 67.17 / 62.66 | 68.83 / 64.67 | 70.72 / 67.32 | **73.23 / 69.55** |
| | FOV Lost | 45.66 / 26.32 | 47.89 / 24.63 | **48.65 / 27.17** | 43.41 / 20.78 |
| | Gaussian (L) | 64.10 / 58.94 | 62.32 / 55.10 | 65.99 / 60.64 | **69.04 / 63.51** |
| | Uniform (L) | 67.28 / 63.21 | 68.68 / 64.72 | 70.18 / 66.81 | **72.54 / 68.79** |
| | Impulse (L) | 67.47 / 63.42 | 69.06 / 65.51 | 70.63 / 67.54 | **72.75 / 68.91** |
| | Gussian (C) | 62.92 / 54.96 | 68.94 / 64.52 | 69.35 / 64.44 | **71.55 / 66.16** |
| | Uniform (C) | 64.43 / 57.61 | 69.33 / 65.26 | 70.06 / 65.81 | **72.46 / 67.99** |
| | Impulse (C) | 63.07 / 55.16 | 68.89 / 64.37 | 69.25 / 64.30 | **71.66 / 66.41** |
| Motion | Compensation | **39.62 / 31.87** | 25.69 / 9.01 | 36.76 / 27.57 | 32.51 / 15.99 |
| | Moving Obj. | 56.41 / 45.43 | 60.03 / 51.01 | 59.42 / 51.63 | **68.12 / 60.62** |
| | Motion Blur | 63.44 / 55.99 | 68.85 / 64.39 | 69.38 / 64.74 | **70.58 / 65.95** |
| Object | Local Density | 67.62 / 63.60 | 69.34 / 65.65 | 70.77 / 67.42 | **72.48 / 68.87** |
| | Local Cutout | 66.45 / 61.85 | 67.97 / 63.33 | 68.11 / 63.41 | **69.62 / 64.17** |
| | Local Gaussian | 66.85 / 62.94 | 67.96 / 63.76 | 68.32 / 64.34 | **71.32 / 67.14** |
| | Local Uniform | 67.92 / 64.09 | 69.67 / 66.20 | 70.68 / **67.58** | **71.34** / 66.03 |
| | Local Impulse | 67.89 / 64.02 | 69.64 / 66.29 | 70.93 / 67.91 | **71.83 / 68.15** |
| | Shear | 61.15 / 55.42 | 66.43 / 62.32 | 62.95 / 60.72 | **68.41 / 65.23** |
| | Scale | 62.00 / 56.79 | 67.81 / 64.13 | 66.00 / 64.57 | **71.40 / 68.90** |
| | Rotation | 63.67 / 59.64 | 67.42 / 63.36 | 66.31 / 65.13 | **71.62 / 68.35** |
| Alignment | Spatial | 67.75 / 63.77 | 69.72 / 66.22 | 71.35 / 68.39 | **71.95 / 69.52** |
| | Temporal | 57.91 / 51.43 | 54.23 / 43.65 | 56.62 / 49.02 | **62.53 / 55.24** |
| Average | | 62.82 / 56.99 | 64.71 / 58.73 | 66.06 / 61.03 | **68.67 / 63.07** |

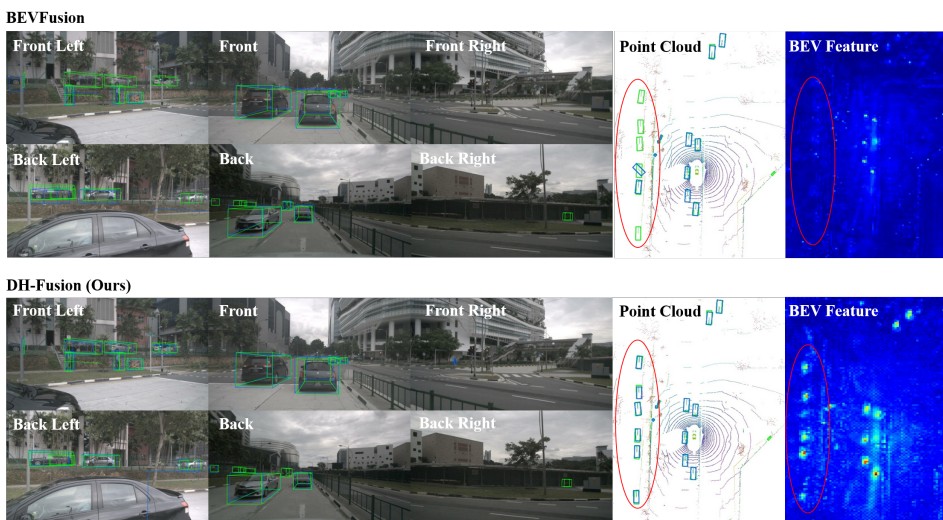

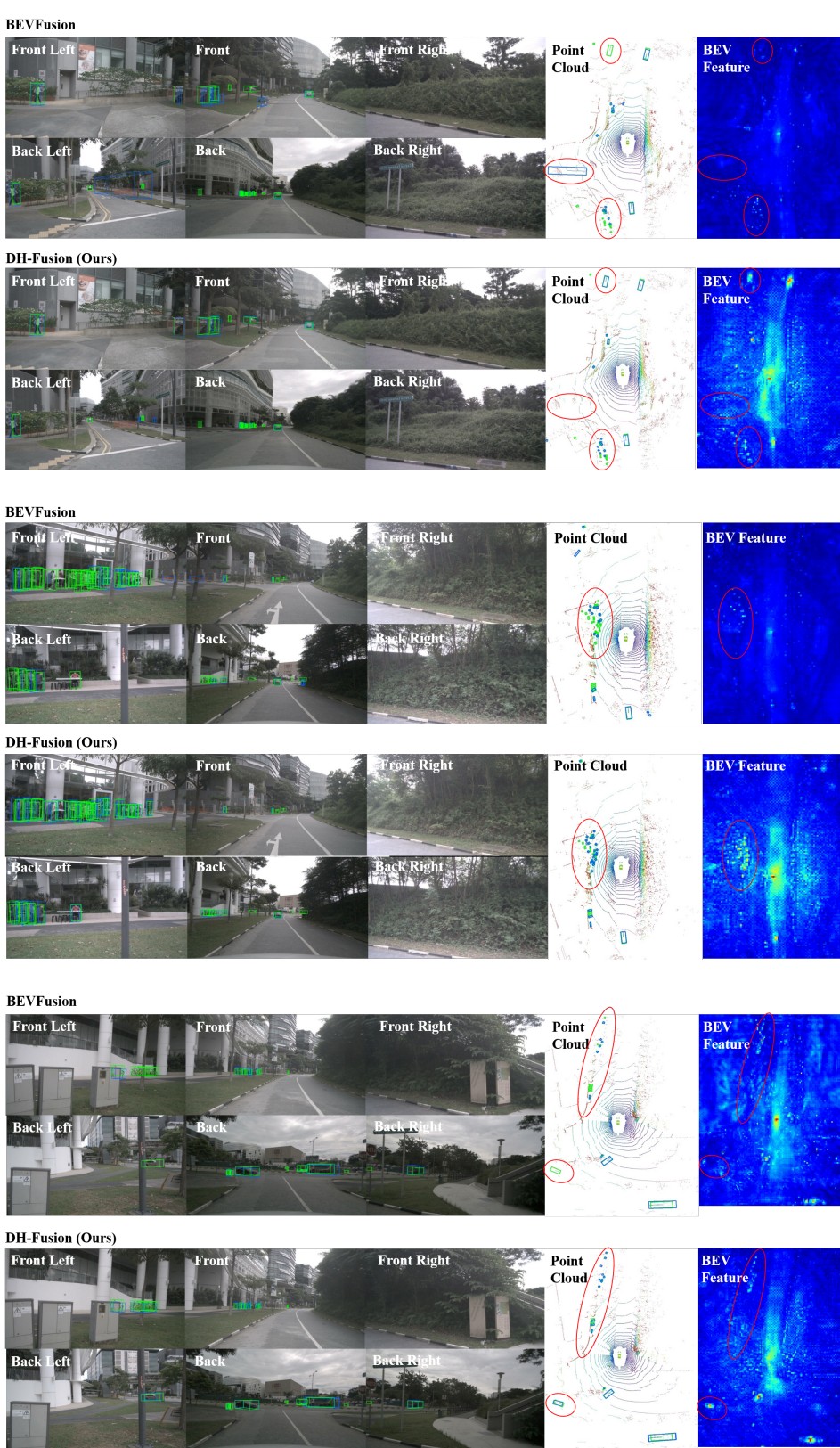

Figure 8: More examples of 3D object detection results and BEV features from BEVFusion and ours. We show the ground truth boxes in green, and the prediction boxes in blue. We use red circles to highlight the comparisons of ours with BEVFusion.

