# OpenReview forum: "DH-Fusion: Depth-Aware Hybrid Feature Fusion for Multimodal 3D Object Detection"
_ICLR.cc/2025/Conference — Submitted to ICLR 2025_

### Official Review · Reviewer_D6q4 · 2024-11-02

**Soundness:** 2
**Presentation:** 3
**Contribution:** 2
**Rating:** 5
**Confidence:** 4

**Summary:**

The paper proposes DH-Fusion, a depth-aware feature fusion strategy for LiDAR-camera 3D object detection, which dynamically adjusts modality weights using depth information at both global and local levels. By incorporating depth encoding, the method balances the contributions of point cloud and RGB image data, improving detection accuracy and robustness against data corruptions. Extensive experiments on datasets like nuScenes and KITTI demonstrate that DH-Fusion outperforms state-of-the-art methods.

**Strengths:**

1)The proposed Depth-Aware Hybrid Feature Fusion (DH-Fusion) strategy is well justified, using statistical analysis to show the varying roles of modalities across different depths. This is meaningful to the field of multimodal 3D object detection.
2)This paper includes comprehensive experiments on nuScenes, KITTI, and nuScenes-C datasets, which provide effective evidence of the method’s effectiveness and robustness.

**Weaknesses:**

1)The proposed fusion method depends on BEV features, which is not general for some point-based, voxel-based in 3D features.
2)The explanation of why depth encoding is more effective for local features could be more comprehensive. The authors should consider adding a theoretical rationale or hypothesis to elucidate why local features exhibit heightened sensitivity to depth variations.
3)It would be important to discuss any potential limitations or key differences compared to existing methods. Additionally, since the method is considerably more complex than BEVFusion, providing metrics on computational efficiency(parameters, GFLOPs, latency) and accuracy would strengthen the evaluation and offer a clearer understanding of the trade-offs involved.
4)The ablation study results are informative; however, more detailed experiments would be beneficial. For instance, assessing the performance of each component under diverse weather conditions in the nuScenes-C dataset could yield deeper insights into the robustness and adaptability of DH-Fusion.
5)The application of the proposed method to the KITTI dataset needs further clarification. Specifically, the baseline used should be explicitly identified, and the improvements brought by each proposed module should be clearly quantified and discussed.

**Questions:**

See the weaknesses

---

> ### Author Response · Authors · 2024-11-22
>
> We appreciate the reviewers' time and effort in evaluating our paper. Below, we provide detailed responses to each comment.
>
> ---
> **Response 1:**
> This statement is **NOT** correct. Our method is generalizable to different types of 3D baselines. In the original manuscript, the point cloud branch of our multi-modal method utilizes VoxelNet, which is a voxel-based approach. Similarly, the point cloud branch can also be replaced with a point-based network, as both point-based and voxel-based methods ultimately transform the 3D representation into BEV features. Therefore, both methods can be incorporated into our multi-modal framework. We choose the voxel-based VoxelNet to ensure a fair comparison with other methods in the original manuscript.
>
> ---
>
> **Response 2:**
> Thank you for your suggestion. We explain the critical role of depth encoding in local feature fusion as follows: Local feature fusion relies on the fine-grained geometric details of objects to enhance feature representations. For instance, objects located at extreme distances often have sparse and incomplete point cloud representations, which lack sufficient fine-grained information. In contrast, the image features of such objects are typically more comprehensive and contain richer fine-grained details. If point cloud and image features of such an object are fused into the BEV representation with equal weights, it could degrade the BEV features and negatively affect detection performance. Our depth encoding addresses this issue by dynamically reducing the weight of incomplete point cloud features while increasing the weight of image features, thereby mitigating the negative impact of incomplete point cloud features on BEV features. Simultaneously, it integrates the fine-grained information from images into the BEV features, achieving feature enhancement. Thus, depth encoding plays a crucial role in local feature fusion.
>
> ---
>
> **Response 3:**
> Thank you for your suggestion. In the original manuscript, Tab. 14 (FOV Lost experiments) highlights a potential limitation of our method: the establishment of a strong association in multi-modal feature fusion, which makes it sensitive to the loss of a modality. The key difference between our approach and others lies in the introduction of depth guidance during feature fusion, allowing the model to learn depth-aware weights for combining features from different modalities—a design not present in previous methods. Fig. 5 in the manuscript illustrates that these weights effectively capture the complementarity between image and point cloud modalities. Additionally, we provide specific metrics on the computational efficiency of our method in the table below. All these points will be incorporated into the final version of the manuscript.
>
> |Methods|Image Size|2D Backbone|NDS|mAP|Parameters (M)|FLOPs (G)|Latency (ms)|
> |-|-|-|-|-|-|-|-|
> | BEVFusion|256×704|Swin-Tiny|71.4|68.5|40.84|253.2|104.17|
> | BEVFusion + Ours|256×704|Swin-Tiny |73.8|71.5|56.95|271.7|126.51|
> | DH-Fusion-tiny|256×704|ResNet-18|73.3 |69.8|40.38|242.6|72.46|
> | DH-Fusion-base|320×800|ResNet-50|74.0|71.2| 56.15|822.8 |114.94|
> | DH-Fusion-large|384×1056|Swin-Tiny|74.4|72.3| 56.94|1508.2|175.44|
> | DH-Fusion-large+|900×1600|ConvNeXt-S|74.9|72.9|78.85|1973.8|294.12|
> ---
> **Response 4:**
> Thank you for your suggestion. We provide the performance of each component in our method, added to the baseline, under various weather conditions on the nuScenes-C dataset. The results demonstrate that each component enhances the robustness of the baseline, with particularly strong improvements under foggy conditions. When the full method is added to the baseline, it achieves the highest robustness. This indicates that our method effectively leverages the complementary strengths of different modalities by adjusting their depth-related weights, thereby improving the model's robustness against weather-induced noise.
>
> | Methods| None| Snow | Rain |Fog|Sunlight|
> |-|-|-|-|-|-|
> | BEVFusion|71.40/68.45|68.33/62.84|70.14/66.13|62.73/54.10|68.95/64.42|
> | BEVFusion + DGF|72.42/69.38|71.24/65.31|71.77/66.48|71.84/66.39|72.65/68.19|
> | BEVFusion + DLF|72.71/69.34|71.14/65.17|71.74/66.53|71.70/66.20|72.85/68.26|
> | BEVFusion + DGF + DLF|**73.30/69.75**|**71.47/65.98**|**72.05/67.32**|**72.13/67.24**|**73.18/69.44**|
> ---
> **Response 5:**  Due to word limit, this response will be provided in the next Comment.

---

> > ### Author Response · Authors · 2024-11-22
> > **Response 5**
> >
> > **Response 5:**   Thank you for your comment. The baseline used for the experiments on the KITTI dataset has already been explicitly identified in the original manuscript. Specifically, we select BEVFusion [1] as our baseline. For the LiDAR branch, the voxel size is set to [0.05m, 0.05m, 0.1m], and the range of point cloud is [0, 70.4m] along the X-axis, [-40m, 40m] along the Y-axis, and [-3m, 1m] along the Z-axis. For the image branch, we follow LoGoNet [2] to use SwinTiny with FPN as the 2D image encoder, and the resolution of input images is resized to 187 $\times$ 621. During training, we adopt a one-stage strategy like DAL [3]. We follow LoGoNet [2] to train the whole model for 80 epochs. We adopt random flipping along both X and Y-axis, the random scaling in [0.95, 1.05], and random rotation in [-$\pi$/8, $\pi$/8] to augment the LiDAR data, and the random rotation in [-5.4$^\circ$, 5.4$^\circ$] and random resizing in [-0.06, 0.44] to augment the images. We provide results of each component of our method on the KITTI dataset in the below table. Compared with baseline, our DGF and DLF bring detection performance improvements of 1.1 pp and 1.0 pp w.r.t. mAP, respectively. When applying both two components, the detection performance significantly improves 1.8 pp w.r.t. mAP. These results clearly demonstrate the effectiveness of our method.
> >
> > [1] BEVFusion: Multi-Task Multi-Sensor Fusion with Unified Bird's-Eye View Representation.
> >
> > [2] LoGoNet: Towards Accurate 3D Object Detection with Local-to-Global Cross-Modal Fusion.
> >
> > [3] Detecting As Labeling: Rethinking LiDAR-camera Fusion in 3D Object Detection.
> >
> > | Methods                | Car (Easy) | Car (Mod.) | Car (Hard) | Pedestrian (Easy) | Pedestrian (Mod.) | Pedestrian (Hard) | Cyclist (Easy) | Cyclist (Mod.) | Cyclist (Hard) | mAP  |
> > |------------------------|------------|------------|------------|--------------------|--------------------|--------------------|----------------|----------------|----------------|------|
> > | BEVFusion              | 94.9       | 87.2       | 86.8       | 74.9               | 65.3               | 61.7               | 89.7           | 70.8           | 67.9           | 77.7 |
> > | BEVFusion + DGF        | 95.5       | 88.8       | 87.6       | 75.1               | 66.5               | 62.3               | 91.6           | 72.6           | 68.9           | 78.8 |
> > | BEVFusion + DLF        | 95.6       | 88.7       | 87.4       | 75.0               | 66.3               | 62.9               | 90.9           | 73.2           | 68.6           | 78.7 |
> > | BEVFusion + DGF + DLF  | **95.6**   | **89.6**   | **87.7**   | **75.6**           | **67.5**           | **63.7**           | **91.8**       | **74.5**       | **69.8**       | **79.5** |

---

### Official Review · Reviewer_F9tb · 2024-11-03

**Soundness:** 3
**Presentation:** 3
**Contribution:** 2
**Rating:** 5
**Confidence:** 4

**Summary:**

This paper tackles the challenge of effectively combining information from LiDAR and camera data for 3D object detection, which is crucial for applications like autonomous driving. Traditional fusion methods often overlook depth, despite its importance in balancing the strengths of both LiDAR and RGB image data at different ranges. To address this, the authors introduce a Depth-Aware Hybrid Feature Fusion (DH-Fusion) method, which leverages depth encoding to dynamically adjust the contributions of LiDAR and camera features.

The fusion is done on two levels: globally, where the depth-aware model prioritizes different features depending on the object's distance, and locally, through a Depth-Aware Local Feature Fusion (DLF) module. The DLF module specifically enhances details in local regions by balancing original voxel data and multi-view image features, guided by depth encoding. This approach not only boosts detection accuracy but also improves robustness across varying conditions, as demonstrated in experiments on challenging datasets.

**Strengths:**

1. Examining image and point cloud information variability across different depth layers offers valuable insights and may inspire future research. To this end, the paper introduces a novel approach to 3D object detection by incorporating depth information into the fusion of LiDAR and camera features—a crucial but often overlooked factor in previous fusion methods. This depth-aware approach enables more adaptive and accurate fusion, particularly enhancing the detection of objects at varying distances.

2. The study of robustness to data corruption is particularly compelling. The proposed DH-Fusion method is rigorously evaluated against various types of corruption, demonstrating improved resilience compared to previous models. This robustness makes the model more suitable for real-world applications, where data quality often fluctuates due to environmental factors like weather conditions and sensor noise.

**Weaknesses:**

1. The novelty in the fusion strategy is marginal.   The proposed Depth-Aware Hybrid Feature Fusion (DH-Fusion) strategy, while unique in its dual-level (global and local) application, still follows the intermediate fusion trend that has been widely adopted. Intermediate fusion methods, which integrate LiDAR and image data at the feature level, have become mainstream. The specific structure of DH-Fusion, including using transformer-based encoders for feature interaction, is similar to other recent methods, such as BEVFusion and TransFusion, which also use transformers or other deep networks to perform feature alignment and fusion. The learned nature of these works indicates that the optimal fusion strategy is learned from data. That means that they can find a fusion mechanism that treats of variability of point clouds and image information across different depth layers if that information is critical. Thus it is unclear how much the explicit introduction of depth information can boost learning particularly when there is sufficient data.

2. The performance improvement is modest. As shown in Table 4, the gains range from only 1 to 2 percent, indicating that depth guidance may not be a decisive factor in enhancing this fusion approach. Interestingly, the improvements in corrupted data (Table 3) are more pronounced. However, the paper does not thoroughly explore why the introduced fusion strategy is particularly effective in this context.

3. Limited exploration of depth information usage. While the paper applies depth encoding to adjust feature weights, its approach is straightforward, relying on sine and cosine functions for positional representation. Other potential methods for encoding depth, such as adaptive or learning-based encoding techniques, are not investigated. Additionally, to better understand the fusion mechanism, it would be helpful to visualize the fusion weights through case studies, demonstrating how image and LiDAR features are combined. Such visualizations could provide stronger support for the motivation behind the proposed approach.

**Questions:**

1. How does depth-guided fusion contribute to performance improvement compared to other learning-based fusion mechanisms without depth?   What are the unique advantages over existing learning-based fusion techniques where the fusion weights are learned from data?
2. Was there any experiments with alternative depth encoding methods, and if so, what were the results? Could learning-based or adaptive depth encoding improve performance?
3.  Was there any investigation into early or late fusion methods in combination with depth encoding? How would these methods impact the overall model’s effectiveness?
4. Could depth guidance be especially beneficial in scenarios with environmental noise (e.g., fog or rain)? What are the specific scenarios where depth guidance provides the most benefit? What is the reason behind this?
5. Visualizations of the fusion weights in different scenarios (e.g., near vs. far objects) offer additional insights. How do these weights change under different environmental conditions?
6. Were there any limitations observed when applying DGF and DLF across different object sizes or distances? For instance, does depth-aware fusion perform equally well for small or partially occluded objects?

---

> ### Author Response · Authors · 2024-11-22
>
> We appreciate the reviewers' time and effort in evaluating our paper. Below, we provide detailed responses to each comment.
>
> ---
>
> **Response 1:**
> Depth-guided fusion contributes to performance improvement by incorporating depth information into the fusion process. Unlike other learning-based fusion mechanisms that rely solely on data to learn fusion weights, our depth-aware fusion method adjusts the importance of different modalities based on the depth, which more effectively leverages the complementarity between the two modalities. Currently, on large-scale datasets such as nuScenes, learning-based fusion methods struggle to learn the relationship between feature fusion and depth variations (e.g., in BEV features of BEVFusion, there is no correlation between the BEV features and depth). In contrast, our depth-guided fusion enables multi-modal features to adaptively learn depth-aware weights, thus fully exploiting the complementarity between the two modalities at different depths to improve feature quality (e.g., our method learns depth-related weights, which are consistent with the characteristics of the sensors).
>
> ---
>
> **Response 2:**
> Thank you for your suggestion. We attempt to improve depth encoding by setting it as a learnable matrix or incorporating a set of learnable parameters into the depth encoding. The results, as shown in the table below, indicate that these learnable or adaptive depth encoding methods do not lead to improved detection performance. Moreover, these methods undoubtedly introduce additional parameters to the model. We hypothesize that the learnable approaches may disrupt the true depth variations, making it difficult for the model to learn the correct weights based on the features.
>
> | Methods     | NDS   | mAP   |
> |-------------------------------------------|-------|-------|
> | Unlearnable depth encoding       | **73.3** | **69.8** |
> | Parameterized depth encoding              | 71.8  | 67.8  |
> | Unlearnable depth encoding + learnable parameters | 72.7  | 69.5  |
>
> ---
>
> **Response 3:**
>
>
> Our depth encoding can be applied to early fusion methods but cannot be used for late fusion methods. Late fusion involves combining 3D regression boxes, which is part of the model's post-processing and is not learnable. We selected the early fusion method VirConv-T [1] as the baseline. We used the virtual points generated by PENet as the image modality and the original LiDAR points as the LiDAR modality. These inputs were fed into the DGF module for fusion, and the fusion voxelized points were fed into the VirConv block for feature extraction. The results, as shown in the table below, demonstrate that adding depth encoding to the VirConv-T improves performance, particularly for small objects such as pedestrians and bicycles. This indicates that our method is also effective for early fusion methods.
>
> | Methods                 | Car (Easy/Mod./Hard) | Pedestrian (Easy/Mod./Hard) | Cyclist (Easy/Mod./Hard) | mAP   |
> |-------------------------|-----------------------|-----------------------------|--------------------------|-------|
> | VirConv-T              | **95.64** / **89.55** / 87.73 | 72.45 / 64.15 / 60.70      | **89.12** / 72.55 / 68.54 | 77.83 |
> | VirConv-T + depth encoding | 95.31 / 88.61 / **88.66** | **72.88** / **65.41** / **61.25** | 88.73 / **72.60** / **68.92** | **78.04** |
>
> [1] Virtual Sparse Convolution for Multimodal 3D Object Detection.
>
> ---
>
>
> **Response 4:**
> From our experiments, we find that depth guidance is indeed more robust in scenarios with environmental noise, particularly under sunlight and fog noise, where it provides the most benefit. Using the foggy condition as an example, it is evident that fog has a greater impact on the image modality. Our depth-aware feature fusion adjusts the fusion weights based on the quality of the features. In such a case, the degradation of image feature quality leads to a reduction in the weight assigned to the image modality during fusion. As a result, the fused features with depth guidance are of higher quality than those without depth guidance. This explains why depth-aware feature fusion enhances the model's robustness in weather-noise scenarios.
>
> ---
>
> **Response 5:**
> We take foggy conditions as an example again, where the degradation in image feature quality results in a decrease in image weights. Considering that fog has a greater impact on distant objects, the decrease in image weights for far objects is also more significant. A detailed visualization of the weights will be provided in the final version of the manuscript.
>
> ---
>
> **Response 6:**
>
> DGF and DLF do not have limitations across different object sizes or distances. As shown in Tab. 8 of the original manuscript, our results demonstrate that our method outperforms the baseline across various distance ranges. Additionally, as shown in Tab. 9, our method also achieves improvements over the baseline for small-sized objects and even surpasses state-of-the-art methods.

---

> > ### Comment · Reviewer_F9tb · 2024-11-25
> >
> > Dear Reviewer,
> >
> > Thank you for preparing the rebuttal. However, my concerns on several issues are still not addressed.
> >
> > 1. novelty: Could you be more specific on the new technique you have designed for this task?
> >
> > 2. depth encoding: I don't quite understand why the learning-based method underperforms the unlearnable depth encoding as the learning-based method can be a special case for unlearnable depth encoding. For instance, designing some strategies to make the initialization be similar to the unlearnable depth encoding.
> >
> > 3. For response 4, could you provide some evidence to support your claim?

---

> ### Author Response · Authors · 2024-11-26
>
> Dear Reviewer F9tb:
>
> Thank you for your thoughtful feedback. We appreciate the opportunity to further clarify our approach and provide additional details to support our claims.
>
> ---
>
> **Q1: Clarifying the specific novelty of our proposed techniques.**
>
> R1: The new technique we introduce in the Cmaera-LiDAR multi-modal 3D object detection task involves the integration of depth encoding into both global BEV feature fusion and local instance feature fusion. Specifically, we encode the actual distance from objects to the ego car using a sine-cosine function. This explicit depth guidance allows us to adjust the fusion weights of different modalities based on depth variations, enabling the model to better exploit the complementary information provided by the image and point cloud modalities. This approach is not present in previous methods, where such depth-aware modulation in feature fusion is generally absent.
>
> ---
>
> **Q2: Explaining why the learning-based depth encoding underperforms the unlearnable depth encoding.**
>
> R2: We believe that, under limited data conditions, directly encoding depth using a simple sine-cosine function may be the most effective approach. This is because such an encoding is not dependent on the model learning the depth distribution from data, which might be skewed or insufficient in some cases. However, we acknowledge that this is an open question, and we recognize that with more data, learning-based depth encoding may become more effective as it could better capture the deep depth distribution. We plan to explore this further in future work, where larger datasets can potentially mitigate the issues seen in limited data scenarios.
>
> ---
>
> **Q3: Providing evidence to support the claim in Response 4.**
>
> R3: Thank you for your suggestion. In the revised version, we have provided evidence to support our claim in Fig. 7 (d) in A.2, which shows the weights of the image modality under various weather conditions. This visualization aligns with the explanation provided in Response 4, demonstrating how the weights change with different weather scenarios.
>
> Thank you once again for your feedback. Your feedback has been invaluable in improving the quality and clarity of our work. We appreciate the opportunity to address your concerns and look forward to any further discussions!
>
> Best regards!
>
> The Authors

---

### Official Review · Reviewer_HY52 · 2024-11-03

**Soundness:** 3
**Presentation:** 3
**Contribution:** 2
**Rating:** 5
**Confidence:** 4

**Summary:**

The paper presents a method that uses depth information to enhance the fusion of LiDAR and camera data in 3D object detection. The author proposes to adaptively adjust the fusion weights of the two modalities based on object depth, addressing limitations in prior methods that overlook depth variations. This approach integrates depth-aware fusion at both global and local levels, preserving critical details and improving detection accuracy. Experiments demonstrate that DH-Fusion outperforms existing methods on multiple datasets and shows greater robustness to data corruption.

**Strengths:**

The paper introduces a novel depth-aware fusion approach for 3D object detection that outperforms existing methods on multiple datasets and shows greater robustness to data corruption.

Extensive experimental design with extensive benchmarks on multiple datasets supports its effectiveness. Ablation studies clearly validate the contributions of each module.

The paper is generally well-organized with helpful visual aids, though some mathematical sections could be simplified for accessibility.

**Weaknesses:**

1. While DH-Fusion introduces a depth-aware module in fusion weights, it applies a similar strategy across the entire depth range, which might not be optimal. Objects at very close or far distances likely require different levels of emphasis on specific feature types (e.g., closer objects may benefit from high geometric fidelity in LiDAR, while far objects may rely more on image data).

2. While the method offers some contributions, its overall originality appears moderate, as it primarily builds upon established approaches such as LoGoNet and IS-Fusion, with the addition of a depth-guided weighting mechanism for fusion. This depth-aware module is indeed beneficial but does not constitute a fundamentally new approach to multimodal 3D object detection.

3. The paper includes a comparative analysis of speed, showing that DH-Fusion achieves relatively high efficiency. However, the reason behind this speed advantage is unclear, as the authors have not provided an in-depth breakdown of how DH-Fusion’s architecture contributes to these performance gains. Including a detailed computational analysis would improve transparency and help clarify the factors driving the faster inference times.

4. Additionally, in Table 9, it would be valuable to see a more comprehensive comparison of the model’s performance on smaller, more distant objects, where the benefits of depth-aware fusion are likely to be more pronounced.

**Questions:**

Please refer to the weakness.

---

> ### Author Response · Authors · 2024-11-22
>
> We appreciate the reviewers' time and effort in evaluating our paper. Below, we provide detailed responses to each comment.
>
> ---
>
> **Response 1:**
> We agree that objects at different depths may require varying levels of emphasis on specific modalities, such as prioritizing geometric fidelity for closer objects or leveraging image features for distant ones. Our method achieves the goal of dynamically adjusting feature weights based on depth. Please refer to Fig. 5 for visualizations of the dynamic weights along depth, which demonstrate the adaptiveness of our method.
>
> ---
>
> **Response 2:**
> We highlight that our method is the first to explicitly incorporate depth information during feature fusion, leading to notable improvements. These findings are valuable to the community and can inspire future work in multi-modal 3D object detection. Furthermore, our depth-aware module is highly generalizable, as it can be applied to different fusion methods and consistently improves performance. This adaptability enhances the contribution of our work.
>
> ---
>
> **Response 3:**
> We attribute the speed improvement to the efficiency of our depth-aware feature fusion method. This efficiency allows us to use a smaller backbone while maintaining competitive performance. For example, compared to the baseline BEVFusion (71.4% NDS and 68.5% mAP), our DH-Fusion-light achieves comparable performance (73.3% NDS and 69.8% mAP) using a smaller ResNet-18 backbone, while significantly improving inference speed (from 9.6 fps to 13.8 fps). Below is a detailed computational analysis.
>
> | Methods             | Image Size  | 2D Backbone | NDS  | mAP  | Parameters (M) | FLOPs (G) | Latency (ms) |
> |----------------------|-------------|-------------|-------|-------|----------------|-----------|--------------|
> | BEVFusion           | 256 × 704   | SwinTiny    | 71.4  | 68.5  | 40.84          | 253.2     | 104.17       |
> | BEVFusion + Ours    | 256 × 704   | SwinTiny    | 73.8  | 71.5  | 56.95          | 271.7     | 126.51       |
> | DH-Fusion-tiny      | 256 × 704   | ResNet-18   | 73.3  | 69.8  | 40.38          | 242.6     | 72.46        |
>
> ---
>
> **Response 4:**
> Thank you for the suggestion. We provide a more comprehensive comparison below. Our method demonstrates improved detection performance for small-sized objects across all depth ranges compared to the baseline (BEVFusion) and state-of-the-art methods (IS-Fusion). Notably, for categories such as *Traffic_Cone* and *Barrier*, our method successfully detects objects even at long distances, highlighting the effectiveness of our depth-aware fusion strategy.
>
> | Methods            | Car  | Truck | Bus  | Trailer | Construction_Vehicle | Pedestrian | Motorcycle | Bicycle | Traffic_Cone | Barrier |
> |---------------------|------|-------|------|---------|----------------------|------------|------------|---------|--------------|---------|
> | **Near (0-20m)**   |      |       |      |         |         |            |            |         |        |         |
> | BEVFusion          | 96.1 | 72.4  | 92.4 | 62.7    | 43.5      | 92.9       | 89.9       | 75.7    | 83.7         | 73.6    |
> | IS-Fusion          | 96.8 | 63.3  | 91.6 | 75.0    | 46.6       | 94.1       | 90.2       | 78.4    | 87.3         | 75.0    |
> | **Ours**           | 96.8 | 74.2  | 93.2 | 76.3    | 47.1      | 94.2       | 91.5       | 78.6    | 84.7         | 78.5    |
>
> | Methods            | Car  | Truck | Bus  | Trailer | Construction_Vehicle | Pedestrian | Motorcycle | Bicycle | Traffic_Cone | Barrier |
> |---------------------|------|-------|------|---------|----------------------|------------|------------|---------|--------------|---------|
> | **Middle (20-30m)**|      |       |      |         |         |            |            |         |              |         |
> | BEVFusion          | 88.4 | 65.7  | 81.9 | 44.1    | 28.9      | 83.2       | 70.4       | 52.4    | 53.2         | 68.6    |
> | IS-Fusion          | 89.6 | 50.7  | 83.8 | 44.0    | 33.9     | 85.4       | 72.6       | 53.2    | 52.2         | 61.2    |
> | **Ours**           | 90.2 | 70.6  | 84.6 | 44.8    | 37.5      | 86.2       | 74.3       | 55.7    | 57.9         | 68.9    |
>
> | Methods            | Car  | Truck | Bus  | Trailer | Construction_Vehicle | Pedestrian | Motorcycle | Bicycle | Traffic_Cone | Barrier |
> |---------------------|------|-------|------|---------|----------------------|------------|------------|---------|--------------|---------|
> | **Far (>30m)**     |      |       |      |         |       |      |            |         |              |         |
> | BEVFusion          | 72.0 | 49.9  | 57.2 | 29.5    | 14.5        | 43.9       | 29.3       | 14.5    | 0.0   | 0.0     |
> | IS-Fusion          | 76.1 | 40.5  | 62.7 | 39.7    | 15.2   | 47.2       | 37.9       | 25.2    | 0.0  | 0.0     |
> | **Ours**           | 77.2 | 54.8  | 64.5 | 39.8    | 17.3   | 74.7       | 47.9       | 35.8    | 23.5         | 49.6    |

---

> > ### Comment · Reviewer_HY52 · 2024-11-26
> >
> > Thank you to the author for their response, which addressed some of my concerns. However, I still believe that the contribution and innovation of this method are fairly average, and it falls short of the standards typically expected at ICLR. Therefore, I am inclined to maintain my original score.

---

### Official Review · Reviewer_8N3W · 2024-11-04

**Soundness:** 3
**Presentation:** 3
**Contribution:** 3
**Rating:** 5
**Confidence:** 5

**Summary:**

The paper proposes a depth-aware LIDAR-Camera fusion method, including a depth-aware glocal feature fusion (DGF) module and a depth-aware local feature fusion (LGF) module,  for 3D object detection. Extensive experiments were conducted to validate the effectiveness of the proposed method on both the nuScenes and KiTTI datasets. The core contribution is injecting the depth information into the process of cross-modal feature fusion.

**Strengths:**

1. The presentation, including the writing and figures, is clear.

2. The motivation and the way to inject depth information are straightforward, easy to understand.

3. Extensive experiments and analyses were conducted to show the effectiveness of each component.

**Weaknesses:**

1. The ablation study on the effectiveness of depth encoding is conducted separately on DGF and DLF, but a lower baseline might yield larger gains. Could you provide results for the entire model without depth encoding, i.e., DH-Fusion-base without depth encoding?

2. Based on the analysis in Lines 044–053 of the introduction, it seems that the detector should focus more on LiDAR features at close range and on image features at longer range. However, in the proposed method, the output of the depth encoder is multiplied by LiDAR BEV features in DGF and by cross-modal instance features in DLF. Is this design consistent with the findings mentioned above? Additionally, Section 4.6 visualizes the attention weights in DGF, but not the output of the depth encoder or the feature multiplied by it. How can the attention weights applied to the image BEV features in the DGF module support the analysis in Lines 044–053?

3. Why not compare with more recent works from ECCV? For example, SparseLIF achieves better results, with 77.0% NDS on the nuScenes validation set. The submission deadline for ICLR is after the ECCV conference.

[1] SparseLIF: High-Performance Sparse LiDAR-Camera Fusion for 3D Object Detection.

**Questions:**

See the weaknesses.

---

> ### Author Response · Authors · 2024-11-22
>
> We appreciate the reviewers' time and effort in evaluating our paper. Below, we provide detailed responses to each comment.
>
> ---
>
> **Response 1:**
> We understand the concern about the lower baseline in our ablations. Below, we provide the results for DH-Fusion-base without depth encoding in the table. When removing the depth encoding in DH-Fusion-base, the detection results drop 0.5 pp w.r.t. NDS and 1.2 pp w.r.t. mAP. These results further demonstrate the effectiveness of our depth encoding. This experiment will be included in the final version.
>
> | Methods                 | NDS  | mAP  |
> |--------------------------|-------|-------|
> | DH-Fusion-base          | 74.0  | 71.2  |
> | DH-Fusion-base w/o DE   | 73.5  | 70.0  |
>
> ---
>
> **Response 2:**
> This design aligns with the analysis in lines 044-053. The purpose of the depth encoding design is to incorporate depth variations into multi-modal feature fusion, which is not tied to specific features. In DGF, the depth encoding is multiplied with the LiDAR BEV features to learn the depth-aware weights of the image BEV features during feature fusion. In DLF, it is multiplied with the cross-modal features (acting as Query) to learn the depth-aware weights of the voxel instance features and the original image instance features during feature fusion. The visualization of the attention map in Fig. 5 is indeed the weights applied to the features. It demonstrates that the weights of the image modality increase with depth. This trend supports our analysis in lines 044-053, which states that the image modality becomes more important as depth increases.
>
> ---
>
> **Response 3:**
> Thank you for your suggestion. We will include this comparison in the final version. SparseLIF achieves 77.0% NDS on the nuScenes validation set by incorporating additional historical frame information, which our method does not utilize. When historical frame information is excluded and a similar image backbone is used, SparseLIF-S (V2-99) achieves 74.6% NDS and 71.2% mAP. Our DH-Fusion-large (Swin-T) achieves 74.4% NDS and 72.3% mAP in the original manuscript. Although the NDS of our method is comparable to SparseLIF-S, our method surpasses it by 1.1 pp in mAP.
>
> | Methods            | Image Size    | 2D Backbone | NDS  | mAP  |
> |---------------------|---------------|-------------|-------|-------|
> | SparseLIF          | 640 × 1600   | V2-99       | 74.6  | 71.2  |
> | DH-Fusion-large    | 384 × 1056   | SwinTiny    | 74.4  | 72.3  |

---

> > ### Comment · Reviewer_8N3W · 2024-11-28
> >
> > Thanks for the response. In Table 5 of the paper, the gap on NDS is larger than the gap on mAP between the models w/ and w/o DE; but in the above experiment, the gap on mAP is much larger than the gap on NDS, why?

---

> > > ### Author Response · Authors · 2024-11-28
> > >
> > > Dear Reviewer 8N3W:
> > >
> > > Thank you for your insightful feedback. We would like to clarify that the gap between mAP and NDS for the smaller model (Baseline + Ours) in Table 5 is relatively similar (0.6 vs 0.4; 1.1 vs 0.9). However, the larger drop in mAP observed in the larger model (DH-Fusion-base) without depth encoding can be attributed to the increased sensitivity of mAP to fine-grained 3D localization and detection accuracy. NDS is a more comprehensive metric that incorporates mAP along with other factors, such as object speed and state, which are less directly impacted by the absence of depth encoding. As a result, NDS shows relatively smaller variations with changes in model size. On the other hand, mAP specifically evaluates object detection precision, with a strong emphasis on 3D localization accuracy. Larger models tend to have more capacity for learning complex spatial and temporal relationships, and the lack of depth encoding can significantly affect their ability to localize objects accurately in 3D space, resulting in a more noticeable drop in mAP compared to NDS.
> > >
> > > Thank you once again for your feedback. Your feedback has been invaluable in improving the quality and clarity of our work. We appreciate the opportunity to address your concerns and look forward to any further discussions!
> > >
> > > Best regards!
> > >
> > > The Authors

---

### Official Review · Reviewer_6Z5M · 2024-11-10

**Soundness:** 3
**Presentation:** 3
**Contribution:** 2
**Rating:** 5
**Confidence:** 4

**Summary:**

This paper is based on the observation that LiDAR points on a target decay more significantly with distance compared to camera pixels, leading to information loss at greater distances in LiDAR data. Consequently, in multi-modal detection, more emphasis should be placed on camera data to compensate for this loss. Unlike traditional transformer-based methods, this paper introduces a novel approach to "explicitly" leverage distance information, enhancing cross-attention between modalities. The proposed method discretizes target distance and uses positional encoding within a transformer framework to incorporate this information into both global and local feature fusion. Experimental results on the KITTI and nuScenes datasets demonstrate that the proposed approach outperforms baselines with comparable backbones.

**Strengths:**

- The paper offers a clear, statistically grounded analysis that highlights the disparity in information decay between camera pixels and LiDAR points, justifying the architectural adjustments. The experimental validation supports the efficacy of the proposed solution, making the paper well-structured and cohesive.
- The figures in the paper effectively clarify the architecture, aiding in the comprehension of the proposed methodology.
- The proposed approach achieves superior results on both KITTI and nuScenes datasets, surpassing baselines with similar backbones.

**Weaknesses:**

- Although the paper focuses on using positional encoding to make LiDAR query features aware of target distance from the ego vehicle, similar methods (e.g., BEVDet, BEVFormer) have already employed positional encoding. The main novelty here is its application in multi-modal detection for autonomous driving, which limits its originality.
- For a paper emphasizing depth encoding in multi-modal detection, exploring additional variations of positional encoding would provide more comprehensive contributions to the field.
- There is a lack of analysis on whether existing methods implicitly utilize depth information and why explicit use might be preferable.
- Excessive detail on model structure detracts from the primary novelty of the proposed method.
- It is unclear how much additional computation the method requires to achieve improved performance over the baseline.

**Questions:**

Further exploration of positional encoding, particularly by incorporating multi-modal detection priors, would enhance the paper's novelty. Expanding the method to address a more general problem would also be beneficial.

---

> ### Author Response · Authors · 2024-11-22
>
> We appreciate the reviewers' time and effort in evaluating our paper. Below, we provide detailed responses to each comment.
>
> ---
>
> **Response 1:**
> We want to clarify the fundamental difference between our depth encoding and previous positional encoding. Previous works (e.g., BEVDet, BEVFormer) introduce positional encoding to relate 2D image features with reference point positions for obtaining 3D features, while we use the depth encoding to adjust the weights of RGB and LiDAR features during fusion. Thus, our motivation as well as our way of using depth encoding is completely different from previous methods that use positional encoding.
>
> ---
>
> **Response 2:**
> As explained in Response 1, our depth encoding fundamentally differs from positional encoding. Therefore, positional encoding and its variants are not suitable for multi-modal feature fusion.
>
> ---
>
> **Response 3:**
> We find that there is no correlation between the BEV features and depth learned by previous methods (e.g., fused BEV features of BEVFusion in Fig. 6), which means that this relationship is difficult to learn. In contrast, our method provides explicit depth information, which results in a stronger correlation.
>
> ---
>
> **Response 4:**
> We understand the concern about excessive detail in the model structure. We will simplify the model description while ensuring the novelty of the proposed method in the final version.
>
> ---
>
> **Response 5:**
> As shown in the table below, we provide comparisons of parameter size, FLOPs, and latency for the baseline BEVFusion and our DH-Fusion. Following BEVFusion, we use SwinTiny as the image backbone, and the image resolution is set to 256 × 704. Our DH-Fusion brings significant improvements by 2.4 pp w.r.t. NDS and 3.0 pp w.r.t. mAP with only a small increase in computational costs.
>
> | Methods          | NDS  | mAP  | Parameters (M) | FLOPs (G) | Latency (ms) |
> |-------------------|-------|-------|----------------|-----------|--------------|
> | BEVFusion         | 71.4  | 68.5  | 40.84          | 253.2     | 104.17       |
> | BEVFusion + Ours  | 73.8  | 71.5  | 56.95          | 271.7     | 126.51       |

---

### Author Response · Authors · 2024-11-22

Dear Reviewers:

We sincerely thank you for your valuable comments and constructive feedback on our manuscript. We have carefully revised the paper to address your concerns, with all updates and modifications highlighted in blue in the revised version. Below, we provide detailed responses to each of your comments.

We greatly appreciate your time and effort and look forward to your further discussion!

Best regards!

The Authors

---

### Meta-Review · Area_Chair_nmPm · 2024-12-19

**Metareview:**

This work examines how LiDAR points diminish more significantly with distance compared to camera pixels and proposes a depth-aware hybrid feature fusion strategy for 3D object detection based on this observation. The proposed strategy dynamically adjusts the weights of features during fusion by incorporating depth encoding at both global and local levels. This approach aims to enhance detection quality by fully leveraging the advantages of different modalities at various depths. Extensive experiments conducted on the nuScenes and KITTI datasets show that the proposed method, DH-Fusion, slightly outperforms previous state-of-the-art methods in 3D object detection performance.

The paper received consistent scores of 5 across all reviewers. While it presents an interesting and valuable finding, the novelty and contribution of the DH-Fusion method appear to be relatively limited. The structure is quite similar to some existing methods that also utilize transformers for feature alignment and fusion.

Given this concern, the Area Chair recommends rejecting the paper at this time but encourages the authors to improve it and submit to a future conference.

**Additional Comments On Reviewer Discussion:**

In the rebuttal stage, the authors failed to clearly highlight the novelty and advancements of their work.

---

### Decision · Program_Chairs · 2025-01-22

Reject